# Room temperature nondestructive encapsulation via self-crosslinked fluorosilicone polymer enables damp heat-stable sustainable perovskite solar cells

Tong Wang [1], Jiabao Yang [1], Qi Cao [1], Xingyu Pu [1], Yuke Li [2], Hui Chen [1], Junsong Zhao [1], Yixin Zhang [1], Xingyuan Chen [1] & Xuanhua Li [1] ✉

Encapsulation engineering is an effective strategy to improve the stability of perovskite solar cells. However, current encapsulation materials are not suitable for lead-based devices because of their complex encapsulation processes, poor thermal management, and inefficient lead leakage suppression. In this work, we design a self-crosslinked fluorosilicone polymer gel, achieving nondestructive encapsulation at room temperature. Moreover, the proposed encapsulation strategy effectively promotes heat transfer and mitigates the potential impact of heat accumulation. As a result, the encapsulated devices maintain 98% of the normalized power conversion efficiency after 1000 h in the damp heat test and retain 95% of the normalized efficiency after 220 cycles in the thermal cycling test, satisfying the requirements of the International Electrotechnical Commission 61215 standard. The encapsulated devices also exhibit excellent lead leakage inhibition rates, 99% in the rain test and 98% in the immersion test, owing to excellent glass protection and strong coordination interaction. Our strategy provides a universal and integrated solution for achieving efficient, stable, and sustainable perovskite photovoltaics.

Perovskite solar cells (PSCs) are being commercialized owing to their high power conversion efficiency (PCE)[1,2] and simple fabrication processes[3]. However, perovskite materials are prone to decomposition under humid air[4], elevated temperatures[5], and continuous illumination[6]. Therefore, the intrinsic instability of perovskite materials severely limits the practical applications of PSCs[7]. Numerous methods such as additive engineering[8–10] and interface engineering[11–13] have been demonstrated to improve the stability of PSCs. However, satisfactory stability cannot be achieved using these strategies alone due to the complex and harsh external environment.

Recently, encapsulation engineering has shown encouraging progress in improving the stability of PSCs based on the knowledge base drawn from the encapsulation technology of commercialized silicon solar cells[14,15]. Polymer sealants based on ethylene-vinyl acetate (EVA)[16], polyisobutylene (PIB)[17,18], Surlyn[19,20], polyurethane (PU)[4,16,21], polyolefin (POE)[16,22], and UV-curable resin[23,24] have been demonstrated to encapsulate PSCs. Although encapsulation can tremendously increase device stability, the encapsulation process such as vacuum hot pressing may damage device efficiency[20,25]. UV resin encapsulation has the advantage of simple and efficient processing, but organic vapors generated during encapsulation may also adversely affect device performance[26]. Chen et al. used paraffin to encapsulate PSCs and realized solvent-free low-temperature processing required for commercialization[27]. However, the encapsulant may cause heat accumulation within the encapsulated

[1]State Key Laboratory of Solidification Processing, Center for Nano Energy Materials, School of Materials Science and Engineering, Northwestern Polytechnical University, 710072 Xi'an, China. [2]Department of Chemistry and Centre for Scientific Modeling and Computation, Chinese University of Hong Kong, Shatin, Hong Kong, China. ✉e-mail: lixh32@nwpu.edu.cn

PSCs because of low thermal conductivity and deteriorating device stability under extreme conditions. Moreover, lead-based PSCs suffer from lead leakage, causing environmental[28,29] and health issues[30,31]. Epoxy resin[32], graphene aerogel[33], and cation-exchange resin[34] have been proven to be effective strategies to prevent lead leakage. However, these strategies focus on suppressing lead leakage, while device efficiency and stability cannot be guaranteed. Therefore, it is imperative to develop the encapsulation materials for achieving the long-term stable operation of high-efficiency perovskite photovoltaics under the framework of sustainability[35].

In this work, we synthesized a crosslinked fluoropropyl methylsiloxane-dimethylsiloxane multiblock polymer (CFDP) with good transparency, UV stability, water and oxygen barrier properties, good adhesion, and thermal stability. The PCE of the encapsulated device was consistent with that of the unencapsulated device, indicating room-temperature nondestructive encapsulation. Furthermore, the target encapsulation exhibited better heat dissipation capacity compared with the typical UV resin encapsulation, effectively avoiding the degradation of device performance due to heat accumulation. The resulting devices maintained 98% and 95% of their initial efficiency under the damp heat and thermal cycling tests, respectively, fulfilling the IEC 61215 standard for silicon solar cells. Moreover, the efficient protection of the glass cover by the flexible encapsulation layer and the strong coordination interaction of C=O groups with $Pb^{2+}$ substantially suppressed lead leakage from shattered devices, following the rain and immersion tests. Our target encapsulation strategy achieves the room temperature nondestructive encapsulation of PSCs, which can optimize thermal management, enhance device stability, and suppress lead leakage.

## Results

### Polymer synthesis and characterization

The CFDP was obtained via the room temperature self-crosslinking of the fluoropropyl methylsiloxane-dimethylsiloxane multiblock polymer (FDP), as shown in Fig. 1a. Supplementary Fig. 1 depicts the synthetic route of the FDP. Fourier-transform infrared spectroscopy (FTIR; Supplementary Fig. 2), Nuclear magnetic resonance spectroscopy (NMR; Supplementary Fig. 3), and X-ray photoelectron spectroscopy (XPS; Supplementary Fig. 4) results showed the successful synthesis of the FDP. Then, the rapid crosslinking of the FDP at room temperature (25 °C) was achieved by the catalyst dibutyltin dilaurate (DBTDL). As shown in Fig. 1b, the FDP was a transparent polymer with certain fluidity before crosslinking. After introducing DBTDL (1% wt) into the polymer system, the flowing polymer was crosslinked and transformed into a viscous solid (CFDP). The Si-O-Si/Si-O-C peak area ratios increased from 1:2 (FDP) to 2:1 (CFDP) (Supplementary Fig. 5), revealing that the Si-O-C in the FDP was transformed into Si-O-Si via the condensation reaction[36]. A small number of outgas were produced during the reaction (Supplementary Fig. 2b).

We measured the light transmittance of the FDP and CFDP deposited on the ITO glass using ultraviolet-visible (UV-vis) spectroscopy, as shown in Fig. 1c. The glass substrates with FDP and CFDP layers had similar light transmittance (from 300 to 850 nm) compared with the original ITO glass. Therefore, the CFDP can be deposited on the front light-incident surface of glass without affecting the light-harvesting efficiency of PSCs. Furthermore, the UV stability of the FDP and the CFDP was investigated by exposing them to UV radiation with a wavelength range of 280–360 nm and an intensity of 120 W m⁻² for 800 h (Fig. 1d). The total exposed UV dose was found to be 96 kWh m⁻², which is more than the permitted exposure dose of 50 kWh m⁻² (IEC 62108 UV conditioning test)[22]. Both FDP and CFDP films exhibited excellent UV stability, and the transmission loss after exposure was less than 1%.

Moreover, we conducted water contact angle, adhesion strength, and thermal performance measurements to reveal the superiority of the CFDP. The water contact angles of the FDP and the CFDP were monitored to evaluate their waterproof ability (Fig. 1e). The FDP and CFDP films had water contact angles of 77.45° and 121.29°, respectively, demonstrating that the CFDP film was more hydrophobic than the FDP film. The water contact angle of the FDP film decreased to 43.69° after exposure to water for 7 min. The intrusion of water might have decreased hydrophobicity. Remarkably, the water contact angle of the CFDP maintained a high value (114.89°), showing good hydrophobicity. The in situ water contact angle tracking (Supplementary Figs. 6 and 7) and lower water vapor transmission rate (Supplementary Table 1) showed that the CFDP not only has a higher waterproof ability but can also suppress water invasion, providing effective moisture and oxygen barrier even in the ambient environment[27]. Furthermore, we compared the adhesion properties of the FDP and the CFDP via the single-lap tensile shear test (Supplementary Fig. 8)[37]. The adhesion strength of the CFDP (0.27 MPa) was higher than that of the FDP (0.05 MPa) in Fig. 1f, indicating that the CFDP has excellent adhesion compared with FDP. In addition, CFDP showed excellent hydrophobicity and strong adhesion strength compared with other encapsulants (Supplementary Figs. 9–11). We further employed thermogravimetric analysis to characterize the thermal properties of the FDP and the CFDP (Fig. 1g). The thermal decomposition temperature (the temperature at 5% mass loss) increased from 301.11 °C (FDP) to 484.28 °C (CFDP), indicating the excellent thermal stability of the CFDP. Therefore, the CFDP shows excellent light transmittance, UV stability, water and oxygen barrier properties, strong adhesion, and good thermal stability, making it a suitable encapsulated material for PSCs.

### Advantages of the target encapsulation strategy

The advantages of using the CFDP as an encapsulant are mainly concentrated in two aspects. First, the CFDP shows better compatibility with perovskite films (Supplementary Fig. 12). More robust emission peak intensity (Supplementary Fig. 13) compared with UV resin is observed, indicating the nondestructive feature of our encapsulation strategy. Further, CFDP exhibits high thermal conductivity (Supplementary Table 2), thus eliminating the potential risk of perovskite decomposition due to heat accumulation. We thoroughly investigated the significance of thermal management for the encapsulation of perovskite. To further improve the thermal conductivity of the CFDP, a small amount of boron nitride was incorporated into the polymer gel to construct the composite gel for target encapsulation[38]. All samples were heated on a hot plate until all samples reached approximately 90 °C, and then quickly transferred to a platform at 25 °C[39]. We used an infrared camera to track the encapsulated perovskite films (Fig. 2a) and obtain the temperature change shown in Fig. 2b. In addition, the POE encapsulation was also compared (Supplementary Fig. 14). The CFDP encapsulation and target encapsulation had a better heat dissipation capacity compared with UV resin encapsulation and POE encapsulation. These results reveal that the composite gel serves as the heat dissipation layer for the target encapsulation strategy, greatly facilitating the heat dissipation of encapsulated perovskite films.

We measured the thermal conductivities of the samples using a hot-disk technology to identify the mechanism of enhanced heat dissipation ability (Fig. 2c)[40]. At room temperature, the thermal conductivities for UV resin, CFDP, and CFDP composite were 0.15, 0.35, and 0.51 W m⁻¹ K⁻¹, respectively. When the temperature reached 55 °C, the thermal conductivities reduced to 0.12 W m⁻¹ K⁻¹ (UV resin), 0.29 W m⁻¹ K⁻¹ (CFDP), and 0.49 W m⁻¹ K⁻¹ (CFDP composite). At 85 °C, the thermal conductivities further decreased to 0.11 W m⁻¹ K⁻¹ (UV resin), 0.26 W m⁻¹ K⁻¹ (CFDP), and 0.45 W m⁻¹ K⁻¹ (CFDP composite). The thermal conductivity of the CFDP composite was much higher than that of UV resin and CFDP at a given fixed temperature[41]. Therefore, the target encapsulation strategy improves heat transfer capabilities due to the high thermal conductivity of the CFDP composite. Furthermore, we used the infrared camera to monitor the temperature

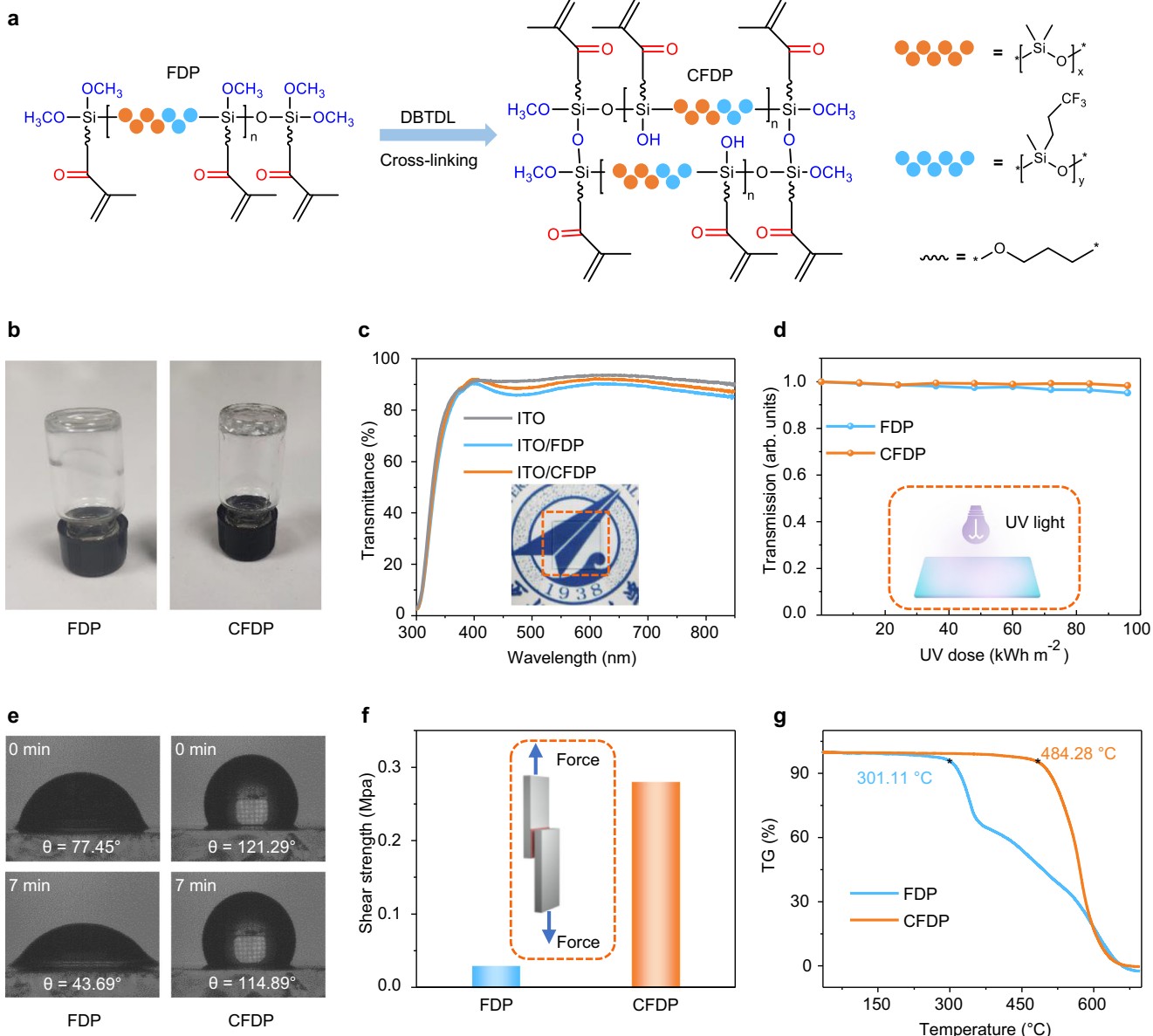

**Fig. 1 | Characterization of FDP and CFDP. a** The synthetic route of polymer gel CFDP. **b** Photographs of the fluorosilicone polymer before (FDP) and after (CFDP) the self-crosslinked process. **c** The light transmittance of ITO glass without and with the FDP and CFDP (thickness of 800 μm). The picture of polymer gel CFDP with a 15 mm-by-15 mm area on an ITO glass substrate is shown in the inset (indicated by orange dashed line). **d** The UV stability of FDP and CFDP. The UV stability test is shown schematically in the inset. **e** The water contact angles of the FDP and CFDP before and after exposure to water for 7 min. **f** The adhesion strength of FDP and CFDP. Inset is a schematic diagram of the bond strength test. **g** Thermogravimetric analysis (TGA) curves of FDP and CFDP. Source data are provided as a Source data file.

of encapsulated samples during continuous heating at 85 °C in Supplementary Fig. 15. After 10 min, the UV resin encapsulated sample still maintained a higher average temperature (83.1 °C) than the target encapsulation (81.2 °C) under thermal equilibrium conditions. Finite element simulation was used to further analyze the heat dissipation capabilities of encapsulated perovskite films (Fig. 2d, e). The corresponding model is presented in Supplementary Fig. 16[42,43]. The perovskite film encapsulated with UV resin exhibited a higher temperature, while the target encapsulation showed good thermal management accompanied by a lower temperature, which agrees well with the infrared images. Therefore, the target encapsulation strategy enhances the ability of heat dissipation efficiency and achieves good thermal management.

The influence of polymer encapsulant on the morphology of perovskite films under heat stress was investigated using scanning electron microscopy (SEM, Fig. 2f, g). We carefully removed the cover glass and encapsulation layer and selected the part of perovskite films without obvious damage for comparison (Supplementary Fig. 17). The perovskite films showed little PbI$_2$ (white phases) before aging. After heat treatment at 85 °C for 500 h, a large amount of PbI$_2$ appeared in the perovskite film with UV resin encapsulation (Fig. 2f), revealing that the heat accumulation might cause the loss of organic components and the decomposition of the perovskite[44]. As a comparison, few PbI$_2$ was observed at the surface of the perovskite film with target encapsulation after aging (Fig. 2g). The cross-linked polymer network loaded with a thermally conductive filler effectively promotes heat transfer and mitigates the potential impact of heat accumulation, which is an important reason for reducing the concentration of PbI$_2$ on the perovskite surface. The X-ray diffraction (XRD) patterns and UV-vis absorption spectra of encapsulated perovskite films are shown in

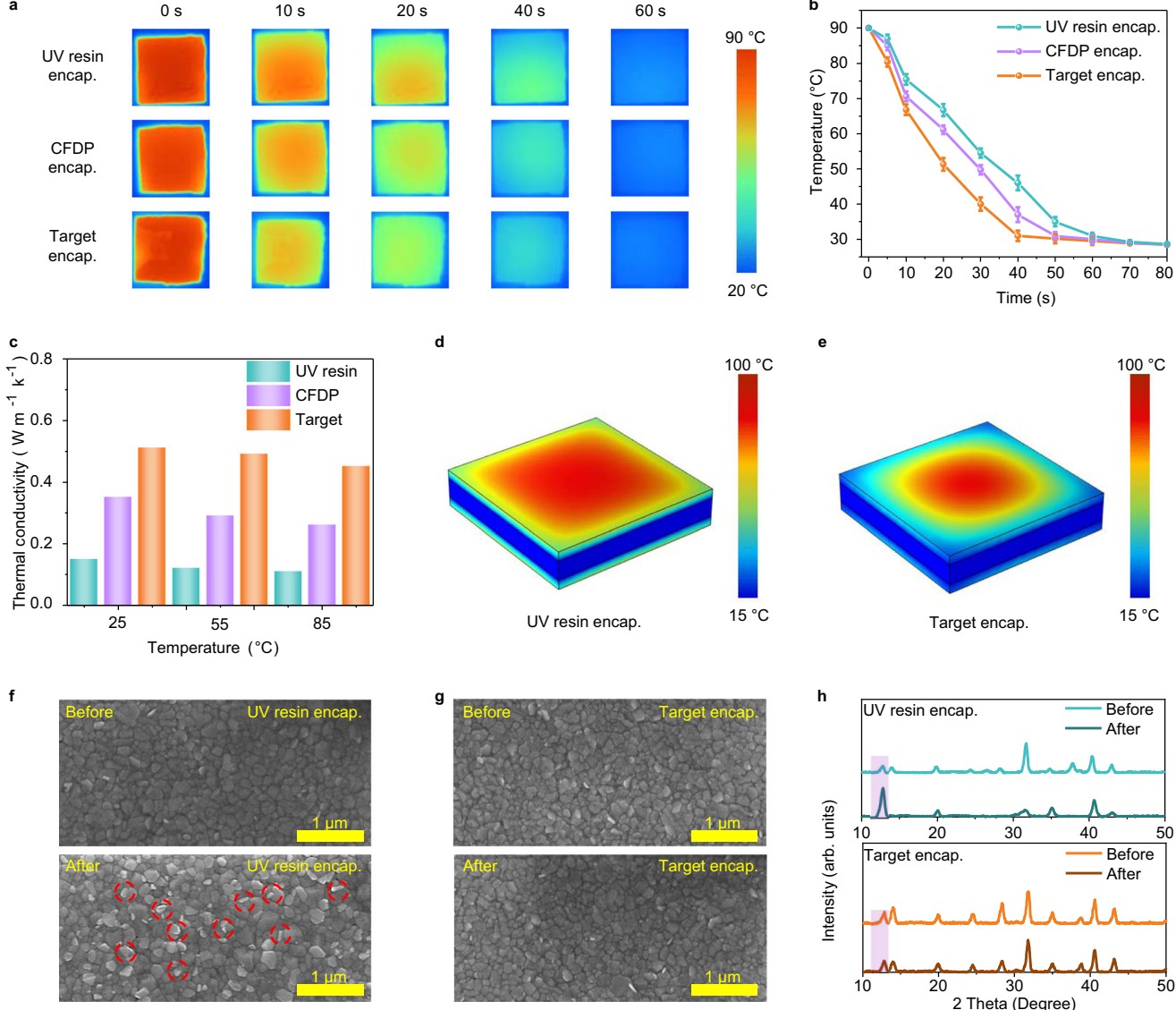

**Fig. 2 | Heat dissipation performance characterization. a** IR thermal images of encapsulated perovskite films during a cooling test. **b** The surface temperature change of encapsulated perovskite films versus time. The error bars represent the standard deviation for three samples. **c** Thermal conductivities of polymer sealant under different temperatures. **d**, **e** The heat transfer model with **d** UV resin encapsulation and **e** target encapsulation in COMSOL Multiphysics. **f**, **g** Top-view SEM images of encapsulated perovskite films with **f** UV resin encapsulation and **g** target encapsulation before and after age in the air at 85 °C for 500 h. Lead iodide is marked with red circles. **h** X-ray diffraction (XRD) patterns of encapsulated perovskite films with UV resin encapsulation and target encapsulation before and after age in the air at 85 °C for 500 h. Source data are provided as a Source data file.

Fig. 2h and Supplementary Fig. 18. Compared with UV resin encapsulation, the perovskite films with target encapsulation showed a weak PbI$_2$ peak and stable light absorption under heat stress. These results show that the proposed encapsulation strategy achieves nondestructive encapsulation at room temperature, promotes heat dissipation, delays perovskite decomposition, and enhances the stability of perovskite films.

**Photovoltaic performance of encapsulated devices**

We fabricated inverted PSCs with a structure of ITO/NiO$_x$/CsMAFA (MA: methylammonium, FA: formamidinium)-based perovskite/ PC$_{61}$BM ([6,6]-phenylC61-butyric acid methyl ester) + C$_{60}$/BCP (bathocuproine)/Cr/Au (Fig. 3a). The schematic diagrams of the PSCs encapsulated with UV resin and target encapsulation are shown in Fig. 3b, c. Supplementary Fig. 19a depicts a schematic diagram of the encapsulation process. The optimum thickness of CFDP encapsulation layer is ~60.25 μm (Supplementary Fig. 19b). The devices

encapsulated with typical UV resin are chosen for comparison (Fig. 3b and Supplementary Fig. 20a, b). The target encapsulation was implemented by cladding around the devices with the CFDP composite, which facilitated heat transfer and inhibited the entry of water and oxygen (Fig. 3c and Supplementary Fig. 20c, d). The control showed a PCE of 22.02% with an open-circuit voltage ($V_{OC}$) of 1.13 V, a short-circuit current density ($J_{SC}$) of 23.81 mA cm$^{-2}$, and a fill factor (FF) of 81.85% (Fig. 3d). The PCE of the device with target encapsulation showed little attenuation and decreased to 21.91% with a $V_{OC}$ of 1.13 V, $J_{SC}$ of 23.72 mA cm$^{-2}$, and FF of 81.75%. The decreased current might have resulted from the slight effect on the optical absorption of the perovskite via encapsulation. By comparison, the device with UV resin encapsulation showed a PCE of 21.44% with a $V_{OC}$ of 1.12 V, $J_{SC}$ of 23.51 mA cm$^{-2}$, and FF of 81.43%. The results reveal that the PCE of the device with target encapsulation showed a slight compromise, satisfying the requirement of nondestructive encapsulation. The statistical results of 15 PSCs showed the reproducibility of the

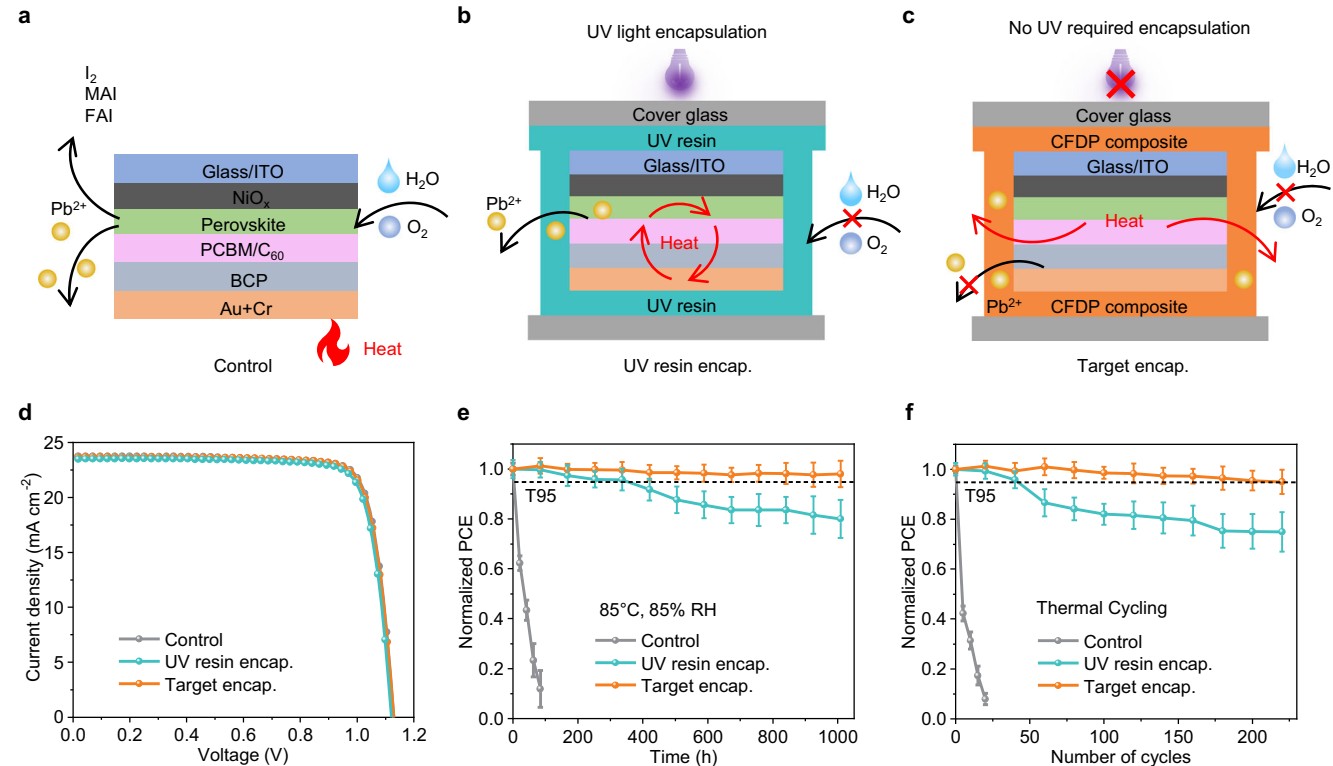

**Fig. 3 | Photovoltaic performance of the PSCs. a–c** Scheme of **a** the unencapsulated PSCs, **b** the PSCs with UV resin encapsulation, and **c** the PSCs with target encapsulation. **d** The current density–voltage (*J-V*) curves of PSCs of the control, UV resin encapsulation, and target encapsulation. **e**, **f** Efficiency evolution of PSCs under **e** damp heat test at 85 °C and 85% relative humidity and **f** thermal cycling test from −40 °C and 85 °C. Error bars represent the standard deviations from the statistic results of three devices. The epoxy resin is used for edge sealing for better stability in all encapsulated methods. Source data are provided as a Source data file.

nondestructive encapsulation strategy with high efficiency (Supplementary Fig. 21).

To estimate the influence of the target encapsulation strategy on device stability, the efficiency evolution was evaluated under various conditions. We tracked the PCE change of the encapsulated PSCs at 85 °C in an N₂ atmosphere (Supplementary Fig. 22 and Supplementary Tables 3 and 4). The results revealed that the devices with target encapsulation decayed more slowly than those with UV resin encapsulation. The devices with target encapsulation also showed better heat dissipation compared with those encapsulated with UV resin (Supplementary Fig. 23). The encapsulated devices were subjected to an environment with 85% RH at 85 °C for 1000 h for the damp heat test according to the IEC 61215 standard (Fig. 3e and Supplementary Fig. 24). The devices with target encapsulation retained 98% of their initial PCE values after 1000 h, while the control device suffered faster attenuation after 100 h under the stress of moisture and temperature extremes. In comparison, the devices with UV resin encapsulation decreased to 80% of their initial PCE. Furthermore, we measured the stability of encapsulated devices under cyclic thermal shocks (Fig. 3f)[45]. The devices with target encapsulation retained 95% of their initial efficiency after 220 cycles, exceeding the requirements of the IEC 61215 standard[17], while the devices with UV encapsulation only maintained 75% of their initial value. In addition, the target encapsulation showed better damp heat stability and thermal cycling stability compared with the POE encapsulation (Supplementary Fig. 25). We further investigated the long-term operational stability of encapsulated PSCs at 55 ± 5 °C under continuous light soaking (Supplementary Fig. 26). The unencapsulated device was fully attenuated after 90 h. The SPO of the device with target encapsulation retained 81% of its original value after 1000 h, while the SPO of the UV resin-encapsulated device was only 52% of the original value. These results suggest that the target encapsulation strategy can endow PSCs with long-term stability.

## Lead leakage under rain and immersion tests

Lead leakage from shattered devices is effectively inhibited by the target encapsulation strategy owing to two main reasons. First, the polymer gel chemically anchors to the glass substrate due to the reaction between the -OH groups on the substrate and the -Si-O-CH₃ groups in the CFDP, which improves the adhesion between the encapsulant and the glass[46]. More importantly, the reaction achieves interfacial toughening and enhances the mechanical reliability of the cover glass[12]. As a result, by using polymer gel with good flexibility and elasticity, a "molecular bridge" can be built between the interface to increase the adhesive strength, achieving a better buffer effect and improving the impact resistance of cover glass. Moreover, the strong coordination interaction of C = O groups in the crosslinked structure with Pb²⁺ is proved in Supplementary Figs. 27–30. Therefore, the excellent protection of the glass substrate and the strong coordination interaction effectively suppresses lead leakage (Fig. 4a).

To further quantify the lead sequestration effect of the target encapsulation, we performed lead leakage measurements. A metal ball was dropped on a glass cover from a constant height to simulate the hail impact, according to the FM 44787 standard testing[47]. After the hail test, we observed that the control device was turned into shards (Fig. 4b). The device with UV resin encapsulation was also seriously damaged, and huge cracks appeared on both sides of the device (Fig. 4c). The device with target encapsulation remained intact with typical star-shaped cracks at the impact location (Fig. 4d). In addition, compared with the POE encapsulation (Supplementary Fig 31), the CFDP encapsulant achieved the better protection for the encapsulated PSCs. We employed a homemade device to simulate the downpour weather after hail (Supplementary Fig. 32)[35]. For structure A, a few portions of the damaged devices changed from black to yellow after several minutes, suggesting that perovskite was decomposed to PbI₂[48]. After 1 h, we observed that the yellow phase extended to most of the

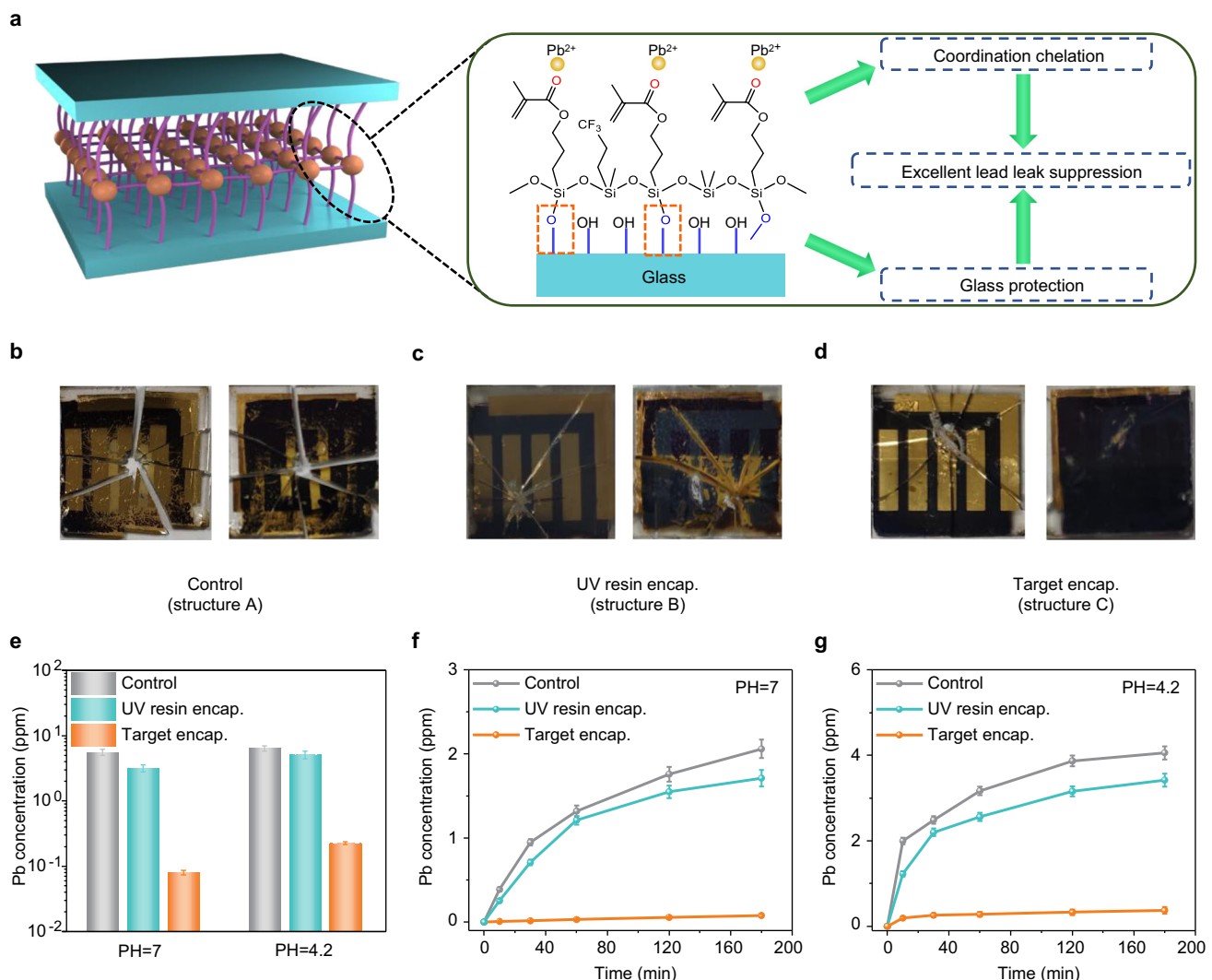

**Fig. 4 | Lead sequestration in the PSCs with target encapsulation. a** Illustration of lead leakage suppression via excellent glass protection and the strong coordination interaction between C = O groups and Pb²⁺. **b**–**d** The typical process for the preparation of damaged PSCs for lead leakage test is based on **b** the unencapsulated PSCs, **c** the PSCs with UV resin encapsulation, and **d** the PSCs with target encapsulation. **e** Water-dripping test results for the damaged PSCs without

encapsulation, with the UV-resin encapsulation, and with target encapsulation. **f**, **g** Water-soaking test results for the damaged PSCs without UV resin encapsulation, and with target encapsulation in **f** neutral water and **g** acidic water. The standard deviations of the three samples are represented by the error bars. Source data are provided as a Source data file.

region (Fig. 4b). For structure B, the yellow phase was limited to the region around star-shaped microcracks (Fig. 4c), while structure C showed a small range in the yellow phase in a smaller region (Fig. 4d). We analyzed the lead concentration via inductively coupled plasma mass spectrometry (ICP-MS), and the calibration accuracy is shown in Supplementary Fig. 33[49]. The average lead concentration in the contaminated water (pH = 7) was $5.59 \pm 0.62$ ppm (structure A) and $3.17 \pm 0.44$ ppm (structure B), and this considerably decreased to $0.08 \pm 0.01$ ppm (structure C) in Fig. 4e, which reached 99% lead sequestration efficiency. To further imitate lead leakage under acidic rain, acidic water (pH = 4.3) was dripped onto the damaged PSCs. Similarly, the average lead concentration reduced from $6.38 \pm 0.60$ ppm (structure A) to $0.22 \pm 0.01$ ppm (structure C), yielding a comparable lead sequestration efficiency of 97%, while the average lead concentration of structure B was $5.12 \pm 0.68$ ppm.

Furthermore, the damaged PSCs were soaked in 40 mL of deionized water to imitate lead leakage in the immersion process (Fig. 4f). In the first 15 min, the concentration of lead seeping from the broken device was quite low, which might be due to the sluggish absorption of water into the perovskite layer. The lead concentration in the

contaminated water from structures A and B increased considerably after 30 min, indicating the UV resin layer and the electrode were no longer capable of preventing water penetration. Remarkably, the device with target encapsulation retained 98% lead sequestration efficiency. We intuitively observed the excellent inhibitory effect of lead leakage from the perovskite films with target encapsulation (Supplementary Fig. 34). We further used the acidic water to soak the damaged PSCs (Fig. 4g). The results showed that the lead concentration of structure C reduced tremendously, which maintained a lead sequestration efficiency of 95%, while structures A and B showed serious lead leakage from the damaged PSCs. Therefore, the target encapsulation strategy efficiently inhibited lead leakage from the damaged PSCs.

## Universality of the target encapsulation strategy

We prepared perovskite devices with diverse perovskite compositions (CsPbI₂Br, MAPbI₃, and CsMAFA) and structures (normal structure, inverted structure, and 25 cm² perovskite module) to illustrate the universality of our encapsulation strategy. The efficiencies of CsPbI₂Br-based devices before and after encapsulation

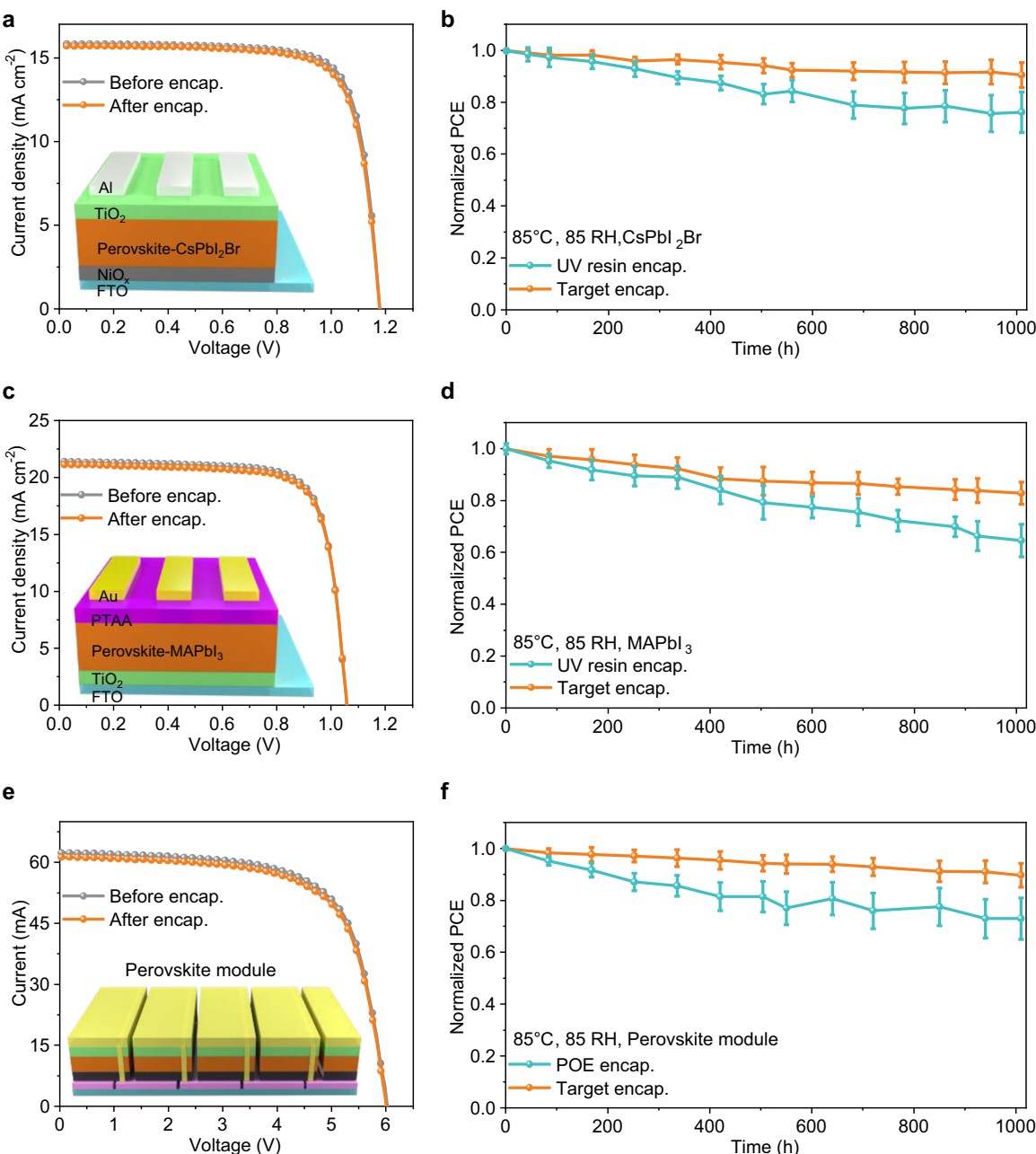

**Fig. 5 | Photovoltaic performance based on the different components and structures. a** The *J-V* curves of CsPbI₂Br perovskite-based inverted devices before and after target encapsulation. **b** The PCE attenuation curve with UV resin and target encapsulation based on the devices of CsPbI₂Br perovskite under damp heat test. **c** The *J-V* curves of MAPbI₃ perovskite-based normal devices before and after target encapsulation. **d** The PCE attenuation curve with UV resin and target encapsulation based on the devices of MAPbI₃ perovskite under damp heat test. **e** The *J-V* curves of CsMAFA perovskite-based modules with 25 cm² before and after target encapsulation (Effective area is 15.8 cm²). **f** The PCE attenuation curve with POE and target encapsulation based on perovskite modules under damp heat test. The standard deviations of the three samples are represented by the error bars. The epoxy resin is used for edge sealing for better stability in all encapsulated methods. Source data are provided as a Source data file.

were similar, as shown in Fig. 5a and Supplementary Table 5, demonstrating that the target strategy achieves efficient encapsulation with minimal damage to the PSCs. We also measured the encapsulated PSCs under the damp heat test (Fig. 5b). The devices with UV resin encapsulation maintained 76% of the initial PCE after 1000 h, whereas the devices with target encapsulation showed slower degradation and retained 91% of the initial PCE. Similar results were observed in the MAPbI₃-based devices, as shown in Fig. 5c and Supplementary Table 6. The devices with target encapsulation maintained 83% of their initial PCE, while the devices with UV resin encapsulation degraded faster and retained only 65% of their initial

PCE (Fig. 5d). In addition, we constructed the normal perovskite module based on CsMAFA perovskite (Fig. 5e, Supplementary Fig. 35 and Supplementary Table 7). As shown in Fig. 5f, the devices with POE (i.e., polyolefin as a typical encapsulant for perovskite module[22]) encapsulation degraded more quickly and retained 73% of the initial PCE, while the devices with target encapsulation retained 90% of the initial PCE. Furthermore, we studied the lead leakage of encapsulated devices based on different structures in the water-dripping tests (Supplementary Fig. 36). All devices with target encapsulation showed excellent lead leakage suppression compared with those encapsulated with UV resin and POE, revealing that the target

encapsulation strategy can be universally used in suppressing lead leakage, regardless of perovskite components, device structures, or active areas. To sum up, the room temperature nondestructive encapsulation strategy exhibits excellent device stability, a simple encapsulation process, and low cost compared with other encapsulation strategies (Supplementary Table 8 and Supplementary Note 1), showing great commercialization advantages.

## Discussion

The synthesized polymer gel CFDP demonstrated good transmittance, UV stability, water and oxygen barrier properties, good adhesion, and thermal stability, making it suitable for the nondestructive encapsulation of PSCs at room temperature. The target encapsulation markedly increased the heat dissipation capability of the encapsulated PSCs and decayed the decomposition of perovskite effectively. The encapsulated CsMAFA perovskite-based devices retained 98% of their initial PCE after 1000 h in the damp heat test and 95% of their initial PCE after 220 cycles in the thermal cycling test. In addition, the polymer gel acted as a buffer layer and filter, achieving an efficient suppression rate under the rain test (99%) and the immersion test (98%) via excellent glass protection and strong coordination interaction between the $C = O$ groups and $Pb^{2+}$. This work offers a customizable encapsulation strategy to achieve durable PSCs with extended lifetimes, optimized thermal management, and suppressed lead leakage, laying the foundation for the sustainable development of PSCs.

## Methods

### Materials

1,3,5-Tris[(3,3,3-trifluoropropyl) methyl] cyclotrisiloxane ($F_3$, 98%) was purchased from TCI. Tetramethylammonium hydroxide (TMAH, 97%), octamethylcyclotetrasiloxane ($D_4$, 98%), dibutyltin dilaurate (DBTDL, 95%), 4-Isopropyl-4′-methyldiphenyliodonium Tetrakis (pentafluorophenyl) borate (TPFB, 98%) and boron nitride (BN, 99.5%) were purchased from Aladdin. 3-(methacryloyloxy) propyltrimethoxysilane (TMPMA, 98%) was purchased from Alfa Aesar. Chloroform (CF, 99.8%), isopropanol (IPA, 99.9%), dimethyl sulfoxide (DMSO, 99.8%), chlorobenzene (CB, 99.8%), $N$, $N$-dimethylformamide (DMF, 99.8%), ethanol (EtOH, 99.9%), titanium tetrachloride (TiCl₄, 99.9%), tetrahydrofuran (THF, 99.9%), and nickel (II) nitrate hexahydrate (Ni(NO₃)₂·6H₂O, 99.99%) were purchased from Sigma-Aldrich. The photovoltaic materials were obtained via Xi'an P-OLED Corp without further purification. The UV resin (LT-U001) was purchased from Luminescence Technology Co., Ltd. The polyolefin (POE) was purchased from First Applied Material Co., Ltd. The synthesis of NiO$_x$ and TiO₂ is detailed in the Supplementary Methods.

### Material synthesis

$F_3$ (37.48 g, 80 mmol) and TMAH (362.4 mg, 2 mmol) were mixed in a dry flask under a nitrogen atmosphere. The mixture was reacted at 40 °C under continuous stirring for 1 h. After adding $D_4$ (23.72 g, 80 mmol), the temperature was raised to 100 °C. After stirring for 2 h, we added the $F_3$ (37.48 g, 80 mmol) at 40 °C. The polymerization reaction was quenched after 1 h with 0.5 mL ethanol. Next, 350 mg of TMPMA (1.4 mmol) was added to 10 g of product that had been dissolved in 5 mL of THF. The reaction was terminated by adding ethanol after stirring for 1 h at 60 °C. Finally, we obtained the colorless viscous liquid FDP after removing THF and ethanol via a vacuum pump. ¹H NMR (400 MHz, CDCl₃): δ 6.03 (s, 1H), 5.48 (s, 1H), 4.05 (t, $J$ = 6.7 Hz, 1H), 3.60−3.46 (m, 3H), 1.99 (dd, $J$ = 15.3, 9.4 Hz, 8H), 1.87 (s, 1H), 0.65 (dt, $J$ = 17.1, 8.0 Hz, 9H), 0.48−0.19 (m, 90H). To obtain the cross-linked polymer gel CFDP, the polymer FDP (1.5 g) and the catalyst DBTDL (15 mg, 1% wt) were added to a glass bottle. After stirring the system at room temperature for 15 min, we obtained the polymer gel CFDP with excellent optical transmittance and adhesion.

### Device fabrication

The ITO glasses (1.5 cm × 1.5 cm) were cleaned with deionized water, acetone, and ethyl alcohol. The substrates were spin-coated with NiO$_x$ (20 mg mL⁻¹) for 30 s at 3000 rpm and heated at 100 °C for 10 min. The perovskite precursor solution (lead bromide (PbBr₂) of 0.21 M, methylammonium bromide (MABr) of 0.21 M, cesium iodide (CsI) of 1 M, lead iodide (PbI₂) of 1.30 M, formamidine iodide (FAI) of 1.19 M in the mixture solution of DMF and DMSO (4/1 V/V), and 1-alkyl-4-amino-1,2,4-triazolium were added into precursor solution as additives) was stirred for 6 h and spin-coated onto NiO$_x$ film at 1000 rpm for 10 s and 6000 rpm for 30 s. The 110 μL of chlorobenzene (CB) was added after 20 s and annealed at 100 °C for 20 min. Then, the PC₆₁BM (20 mg) and C₆₀ (5 mg) were dissolved in 1 mL CB and spin-coated at 5000 rpm for 50 s and annealed at 60 °C for 5 min. Vacuum evaporation equipment was used to evaporate the BCP (8 nm), Cr (4 nm), and Au (100 nm) at $2 \times 10^{-6}$ mbar. Metal mask of 0.1 cm² was used during metal electrode fabrication. The effective area of the device was 0.1 cm². The fabrication of other devices is detailed in the Supplementary Methods.

### Device encapsulation

The encapsulation process involved the application of FDP, adding DBTDL, and waiting for solidification. First, the FDP polymer (1 g) and DBTDL (0.01 g, 1 wt%) were mixed in a glass bottle. We controlled the thickness of the encapsulant via the spin coating and obtained the optimum thickness by adjusting the spin coating speed (2000 rpm). After stirring for 1 min and waiting 3−5 min, the CFDP was spin-coated at 2000 rpm for 30 s on the substrates. After 25 min, the CFDP was completely cured. The encapsulation process was completed before the CFDP was fully cured. We incorporated the boron nitride (2% wt) into the polymer system for the construction of composite gels and spin-coated the polymer film in an N₂ glove box. The structure of encapsulated PSCs was cover glass/CFDP composite/PSCs/CFDP composite/cover glass. For the device stability test, we used epoxy resin for edge sealing. Moreover, the step of UV encapsulation was similar to the target encapsulation. The encapsulated PSCs were uniformly scraped and then cured under UV light for 2 min[21]. The thickness of the UV resin layer was slightly thicker than the CFDP (~60.25 μm), which was ~65.67 μm. Encapsulation of perovskite modules with CFDP composites was similar to the encapsulation steps of PSCs. Encapsulation of perovskite modules with POE film was accomplished by hot-pressing in a vacuum environment. During the encapsulation process, we set the lamination time to 15 min, the pressure to 50 kPa, and the temperature to 120 °C[16].

### Film characterization

FTIR spectroscopy was measured by FTIR-6100 (Jasco). XPS was measured by the Kratos instrument (Axis Supra). NMR spectroscopy was tested by the AVANCE 600 MHz (Bruker). The water contact angles were measured by DSA100 (KRUSS). The UV-vis absorption spectroscopy and light transmittance were obtained by the spectrophotometer U-3310 (HITACHI). The tensile testing equipment (Instron 5942) with a 500 N load cell was used to perform the single-lap tensile shear strength test. The thermogravimetric analysis (TGA) was conducted by the simultaneous thermal analyzer (NETZSCH STA 449F3). The thermal conductivity was obtained by Hot-Disk TPS2200. The water vapor transmission rate of CFDP film (the area was 25 cm² and thickness was 1 mm) was derived from the measurement via the Water Vapor Transmission Rate Test System (C306H) at a selected temperature (25 °C) and humidity (90% RH). The gas chromatography (Shimadzu GC-2014C, molecular sieve-5A, and Ar as the carrier gas) was used to acquire the outgassing release during the reaction process. The encapsulant was cured in vials and heated at 85 °C for 1 h to facilitate outgassing. After 1 h, the sealed gas was sampled by a heated glass 1 mL syringe. We quantified the peak intensity by peak area via the software and calculated the amount of outgassing (11.5 μmol g⁻¹) by

comparing the peak area of the standard gas. SEM images were obtained by SEM (FEI, Verios G4) with a bias of 10 kV. XRD spectroscopy was obtained by XRD-7000 (Kratos, Cu Kα). The photoluminescence (PL) spectra were acquired by the FLS980 spectrometer (Edinburgh).

## Device characterization

The PSCs were conducted by the source meter (Keithley 2420) and solar simulator (Class AAA, 100 mW cm$^{-2}$, Newport 94023 A Oriel Sol3A) under AM 1.5 G illumination. A silicon reference cell (Hamamatsu S1133) was used to calibrate the intensity of light. All J-V results are measured in a nitrogen-filled glove box, using the scanning speed of 0.1 V s$^{-1}$ from -0.1 V to 1.2 V. IR thermal images of encapsulated PSCs were obtained with an Infrared camera (Fluke Ti300). The encapsulated devices were tested for operation stability under continuous one-sun illumination by MPP tracking equipment (YH-VMPP-16) at 55 ± 5 °C. The encapsulated devices were measured at 85 ± 0.5 °C and 85 ± 0.5% RH in the constant temperature and humidity chamber (HS-408L) for the DH tests. The encapsulated devices were placed in the chamber, with the temperature cycling between −40 °C and 85 °C. The temperature change rate between −40 °C and 85 °C was set to not exceed 100 °C h$^{-1}$, and the temperature was stable for at least 15 min at the temperature point of −40 °C and 85 °C, respectively. The thermal stability of the encapsulated PSCs with UV resin encapsulation and target encapsulation was measured by placing them on a hotplate at 85 °C. The encapsulated devices were tested periodically as follows the procedure after the devices cooled down.

## Computational method

Finite element simulation was conducted by building the physical models of heat transfer in the COMSOL Multiphysics. The solid heat transfer model was calculated as the following equation: $\rho C_\rho \mu \triangle T + \triangle q = Q + Q_{ted}$ and $q = -k \triangle T$, where $k$, $C_\rho$, $T$, and $\rho$ represent thermal conductivity, specific heat capacity, temperature, and density. The middle layer represents the perovskite film and the thermal conductivity is 0.5 W m$^{-1}$ K$^{-1}$[50]. The top and bottom films represent the encapsulated polymer film, and their thermal conductivity is measured in the experiment. Gaussian 09 program was used to calculate binding energies. A split-valence def2-SVP basis set and B3LYP method were used to optimize the structures. Van Der Waals interaction was described by DFT-D3 dispersion correction. The binding energy $\Delta E$ was calculated as the following equation: $\triangle E = E_{Pb^{2+}/m} - E_m - E_{Pb^{2+}}$, where $E_{Pb^{2+}/m}$, $E_m$, and $E_{Pb^{2+}}$ represent the energy of Pb$^{2+}$ binding with the molecule, the molecule, and Pb$^{2+}$, respectively. The calculation of electrostatic potential is detailed in the Supplementary Methods.

## Lead leakage characterization

The lead content in the polluted water was determined via ICP-MS (Thermo Fisher Scientific). The standard curve was obtained by configuring standard lead solutions of different concentrations (0, 1, 5, 35, 50, 100 ppm). Lead sequestration efficiency (SQE) was defined by SQE $(\%) = \left(1 - \frac{\text{Pb leakage from encapsulated devices}}{\text{Pb leakage from unencapsulated devices}}\right) \times 100\%$[35].

## Reporting summary

Further information on research design is available in the Nature Portfolio Reporting Summary linked to this article.

## Data availability

All data generated in this study are provided in the article and Supplementary Information, and the raw data generated in this study are provided in the Source Data file. Source data are provided with this paper.

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

## Acknowledgements

This research is supported by the National Natural Science Foundation of China (22261142666, 52172237), the Shaanxi Science Fund for Distinguished Young Scholars (2022JC-21), the Research Fund of the State Key Laboratory of Solidification Processing (NPU), China (Grant No. 2021-QZ-02), and the Fundamental Research Funds for the Central Universities (3102019JC005, D5000220033). All funding is awarded to X.L. We thank the members from the Analytical & Testing Center of Northwestern Polytechnical University for the help of XRD, XPS, and SEM characterization.

## Author contributions

T.W. and X.L. proposed the experimental concepts, designed the experiments, and prepared the manuscript. X.L. supervised the project. J.Y., Q.C., and X.P. contributed to the fabrication of the perovskite solar cell. H.C. and J.Z. carried out the optimization and characterization of the devices/samples. Y.Z. and X.C. conducted the measurement of lead leakage. Y.L. finished the computation.

## Competing interests

The authors declare no competing interests.
