## [Peer Review File · Nature Communications]

Room temperature nondestructive encapsulation via self-crosslinked fluorosilicone polymer enables damp heat-stable sustainable perovskite solar cellsREVIEWER COMMENTS

Reviewer #1 (Remarks to the Author):

Wang et al. reported a self-crosslinked polymer gel for nondestructive room-temperature encapsulation strategy for PSCs. The encapsulated PSCs shows significantly enhanced stability among damp-heat test, thermal cycling test and lead leakage test. The stability results in this manuscript fulfilling the IEC 61215 standard are impressive and instructive for PSCs towards commercialization. Therefore, it is recommended that the manuscript undergoes following revisions before consideration for Nature communication.

The conventional encapsulation with industrial standard needs polymer sealants as inner encapsulants and butyl rubber as edge sealing. The inner polymer sealants can't prevent the moisture and oxygen enough. If the self-crosslinked strategy shows a better moisture and oxygen protection than the typical encapsulation strategy with edge sealing, it would be an impressive result for PV encapsulation, especially for perovskite modules. Do the authors think that such cross-linking strategy is enough even without polymer sealants such as EVA or POE?

The authors claimed the thermal management of the CFDP is optimized while the samples are placed on a hot plate. The thermal environment is different among the hot plate test, damp-heat test, and outdoor operation durability. Only the damp-heat chamber is in a thermal equilibrium condition; therefore, the heat dissipation of the CFDP can't support the damp-heat stability.

The thermal management of the device encapsulated by CFDP would be important for a working device. Because the temperature of a working device would be 50~60°C under continuous light soaking, a good heat transfer is significant for long-term power output. SPO tests at higher temperatures than 45C would be a better option.

The devices after thermal cycling show a faster degradation than those after damp heat testing. Could the authors explain possible reasons for such faster degradation?

In Figure 5, devices with Spiro were used for thermal stability tests. Since Spiro itself is not thermally stable, it is hard to use such a structure to compare the thermal stability of encapsulation materials.

Reviewer #2 (Remarks to the Author):

This manuscript reports a new transparent self-crosslinked polymer (CFDP) as an encapsulant material for Perovskite Solar Cells. The material shows good hydrophobic properties and ability to prevent lead leakage. The stability result is impressive, passing both thermal cycling and damp heat. The author claims that the advantage is its good thermal management ability leading to effective heat dissipation under thermal stress, leading to lower level of perovskite composition. The author should include more in-depth discussion, including but not limited to ability for outgas suppression and mechanical properties enhancement. In addition, the comparison between CFDP and UV resin is not enough to showcase the advantages of CFDP. The author should compare with other reported successful encapsulation (which pass IEC tests, such as PIB/POE) from the literatures as well. Therefore, a revision is required.

Here are my detailed comments:

1. The only alternative encapsulant is UV resin and it is already known that UV resin can damage the device. Others have reported non-transparent polymer PIB or transparent polymer POE for encapsulating perovskite and tandem devices, all showing negligible device performance change (even at higher temperature) and passing the IEC test. Simply comparing between CFDP and UV Resin is not sufficient to showcase the advantages. A more comprehensive analysis on adhesion, heat dissipation, stability with other polymer materials should be presented. Besides, encapsulation process complexity and cost should be compared as well.

2. No details are provided for the deposition of FDP and CFDP layers on glass substrates. Does it involve application of FDP, adding DBTDL, and waiting for solidification? Does it have a similar solidification mechanism of a 2-part epoxy? Is there any outgas from the solidification process from CFDP? The curing of CFDP is not provided in the paper either, such as how long for CFDP to become a solid and what curing conditions are required. These are useful for understanding the encapsulation adhesion mechanisms.

3. What are the thickness of CFDP and FDP layers and how they are compared to other encapsulates such as UV resin?

4. CFDP shows enhanced hydrophobicity and adhesion strength. As CFDP is formed by incorporating DBTDL into FDP, please explain how DBTDL improves the hydrophobicity and adhesion strength. Also, readers are more interested in how CFDP is compared with UV resin and other encapsulants regarding those two properties. FDP is not used as an encapsulant in this paper.

5. In Table S1, the author listed the water vapor transmission rate of CFDP. How is this value derived, from reference or measurement?

6. Please include CFDP-only results from Figure S11-12 in Figure 2. This can give a more straight forward comparison to show the advantage of CFDP over UV resin. Again please compare with other encapsulants.

7. In Figure 2a and 2b, why target encapsulation has a faster cooling rate at the beginning and a slower cooling rate at the end while UV resin cooling rate is quite constant. Why the edge of both samples has a lower temperature even from the beginning?

8. In Figure 2f-h, both SEM and XRD show that after 85 degC thermal stress for 500hour, UV resin encapsulated sample shows more severe perovskite decomposition due to poor heat dissipation ability. However, when the sample is under constant thermal stress, the heat exchange should be different to the previous cooling conditions. Is it possible to directly use the infrared camera to measure the temperature of the samples during thermal stressing. In this way, it may be possible to directly check whether the temperature of UV resin encapsulated sample is much higher all the time, being a more direct evidence to show heat accumulation.

9. Do you think the more significant degradation also comes from weaker outgas suppression ability of UV resin? Please discuss on this considering material density/thickness, adhesion, outgas permeability and etc.

10. On page 8 and in Figure 4, Pb leakage measurement, it is clearly that the breaking of all three samples are different. Is it because the metal ball drops differently or does it mean CFDP has a stronger mechanical property? or is CFDP thicker than UV resin so it can provide a good protection mechanically? The author has mentioned enhanced mechanical strength due to the reaction of -OH in the glass and -Si-O-CH₃ in CFDP. However, what I understand is this reaction can improve the adhesion between the encapsulant and the glass but why the whole device becomes tougher upon damage? Also please compare the mechanical strength of other encapsulants vs CFDP as well.

11. Please state whether the encapsulated perovskite film in this studies have cover glass or not for SEM, XRD, PL, temperature cooling test and lead leakage test. If there is a cover glass, how did you remove the encapsulants and cover glass after stressing to measure SEM, XRD and PL? Will removing the encapsulation damage the film as well?

12. In Figure 5, the second structure is based on MA and Spiro, which is more vulnerable to thermal stress. Why is the stability trend similar to the first inorganic CsPbIBr based devices? The thermal

stabilities of MA and Cs based perovskite as well as NiOx and Spiro are clearly different even when the same encapsulation method is used.

Reviewer #3 (Remarks to the Author):

This manuscript introduced an innovated perovskite encapsulation materials by synthesis CFDP and its composite blends, which enriches the choice for the encapsulants of perovskite solar cells. The authors systematically characterize the validity of the synthetic procedure of CFDP and its intrinsic physical properties. The CFDP composite encapsulated device maintained 98% and 95% of its initial PCE (21.91% after CFDP encapsulation) after 1000h damp heat test and 220 cycles of cyclic thermal shocks respectively, which satisfy the IEC61215:2016 standard. I think the work is interesting and important for the PVSK community, and hence it can be considered for publication on Nature Communication.

Some questions need to be addressed:

1. The authors describe the CFDP as a thermally stable molecule. However, the CFDP sample starts to lose its weight when the temperature only raised to 150 oC while 10% of its initial weight was lost at 505.17 oC, according to the TGA characterization in Fig.1(g). A further explanation is needed about why the crosslinked molecule shows a lower initial degradation temperature with its potential impact.
2. According to the description in line 117, BN was incorporated in the CDFP gel to improve the thermal conductivity. However, the UV resin did not add equivalent BN before UV cured, which makes the optimized thermal dissipation less convective when the data of the intrinsic thermal conductivity of CFDP cannot be found in the manuscript.
3. In Line 199, the author mentioned that the C=O bond in the CFDP molecule could establish strong interaction with the Pb²⁺ in perovskite, which may not correspond well with Fig2. (g). The SEM image above did not show drastic changes when the CFDP was removed from the surface of perovskite. I think the authors should describe the preparation process of the SEM sample, for viscous encapsulant are usually hard to remove when the encapsulation is done. Meanwhile, is it possible that the strong chemical bond formed between Pb²⁺ and C=O could reduce the concentration of PbI₂ on the surface of perovskite layer (PbI₂ removed alone with the removal of CFDP encapsulant)?
4. The encapsulation process with a lamination at 120 oC for 15 min does not show any damage to the performance of the solar cell device in terms of PCE. This is usually difficult for the PVSK device. Does this have something to do with the CFDP itself or the BN added into the encapsulant?

Point-by-point Response to the Reviewers' Comments

Reviewer #1

Wang et al. reported a self-crosslinked polymer gel for nondestructive room-temperature encapsulation strategy for PSCs. The encapsulated PSCs shows significantly enhanced stability among damp-heat test, thermal cycling test and lead leakage test. The stability results in this manuscript fulfilling the IEC 61215 standard are impressive and instructive for PSCs towards commercialization. Therefore, it is recommended that the manuscript undergoes following revisions before consideration for Nature communication.

Response: We deeply appreciate the positive comments of the reviewer.

1. The conventional encapsulation with industrial standard needs polymer sealants as inner encapsulants and butyl rubber as edge sealing. The inner polymer sealants can't prevent the moisture and oxygen enough. If the self-crosslinked strategy shows a better moisture and oxygen protection than the typical encapsulation strategy with edge sealing, it would be an impressing result for PV encapsulation, especially for perovskite modules. Do the authors think that such cross-linking strategy is enough even without polymer sealants such as EVA or POE?

Response: Thanks for the comment. The cross-linking strategy is enough even without polymer sealants such as EVA or POE. In the revised version, we compare the stability of PSCs with different encapsulated methods. As shown in **Figure R1**, we observe that the target encapsulation shows better damp heat stability and thermal cycling stability compared with the UV resin and POE encapsulation.

Furthermore, our cross-linking strategy has many advantages compared with the EVA and POE. First, CFDP shows good transparency, UV stability, water and oxygen barrier properties, good adhesion, and thermal stability. Second, CFDP can achieve non-destructive encapsulation at room temperature, avoiding the high-temperature hot pressing process under the vacuum environment.¹ Third, the composite gel has

excellent thermal conductivity, delaying the degradation of perovskite under thermal stress. Fourth, our encapsulation materials exhibit outstanding lead leakage inhibition rates.

Figure R1 Efficiency evolution of PSCs under (a) damp heat test at 85 °C and 85% relative humidity and (b) thermal cycling test from -40 °C and 85 °C. Error bars represent the statistical results of the three devices. The epoxy resin is used for edge sealing for better stability in all encapsulated methods.

In the revised manuscript, **Figure R1** is added as **Supplementary Figure 25** and the relative discussion has been added on page 9 in the main text (also copied as below).

“In addition, the target encapsulation showed better damp heat stability and thermal cycling stability compared with the POE encapsulation (Supplementary Fig. 25)”

2. The authors claimed the thermal management of the CFDP is optimized while the samples are placed on a hot plate. The thermal environment is different among the hot plate test, damp-heat test, and outdoor operation durability. Only the damp-heat chamber is in a thermal equilibrium condition; therefore, the heat dissipation of the CFDP can't support the damp-heat stability.

Response: Thanks for the suggestion. To reveal the heat dissipation capacity of encapsulated perovskite films under thermal equilibrium conditions, the infrared camera is used to monitor the temperature of the samples during continuous heating at 85 °C in **Figure R2**. After 10 min, the temperature of the UV resin encapsulated sample is higher than that of the target encapsulated sample obviously. The average temperature is 83.1 °C and 81.2 °C for the UV resin and target encapsulated sample, respectively, which shows more heat accumulation for the UV resin encapsulated sample. Therefore,

the target encapsulated sample has a lower temperature and better heat dissipation capability than the UV resin encapsulated sample under thermal equilibrium conditions.

Figure R2 (a) IR thermal images of UV resin and target encapsulated perovskite films under thermal stress. (b) Statistical histogram of the average temperature. The encapsulated sample reaches the thermal equilibrium state when the average temperature reaches the stable value.

In the revised manuscript, **Figure R2** is added as **Supplementary Figure 15** and the relative discussion has been added on page 7 in the main text (also copied as below).

*“Furthermore, we used the infrared camera to monitor the temperature of encapsulated samples during continuous heating at 85 °C in **Supplementary Fig. 15**. After 10 min, the UV resin encapsulated sample still maintained a higher average temperature (83.1 °C) than the target encapsulation (81.2 °C) under thermal equilibrium conditions.”*

3. The thermal management of the device encapsulated by CFDP would be important for a working device. Because the temperature of a working device would be 50~60 °C under continuous light soaking, a good heat transfer is significant for long-term power output. SPO tests at higher temperatures than 45 °C would be a better option.

Response: Thanks for the suggestion. The SPO tests at 55±5 °C have been supplemented in the manuscript in **Figure R3**.

Figure R3. The SPO attenuation curve of control, the device with UV resin encapsulation, and the device with target encapsulation at maximum power point under AM 1.5 illumination at 55 ± 5 °C in the air. The epoxy resin is used for edge sealing for better stability.

In the revised manuscript, **Figure R3** is added as **Supplementary Figure 26** and the relative discussion has been added on page 10 in the main text (also copied as below). “We further investigated the long-term operational stability of encapsulated PSCs at 55 ± 5 °C under continuous light soaking (**Supplementary Fig. 26**). The unencapsulated device was fully attenuated after 90 h. The SPO of the device with target encapsulation retained 81% of its original value after 1000 h, while the SPO of the UV resin encapsulated device was only 52% of the original value.”

4. The devices after thermal cycling show a faster degradation than those after damp heat testing. Could the authors explain possible reasons for such faster degradation?

Response: Thanks for your insightful question. The damp heat stability not only ascertains the effectiveness of the encapsulation against the humidity but also tests the thermal stability of the perovskite itself at 85 °C.² Due to the good water/oxygen barrier properties and excellent heat dissipation capability of our encapsulation strategy, the encapsulated devices show excellent damp heat stability. However, the thermal cycling stability test is challenging due to the linear expansion coefficient mismatch between perovskite and the charge transport layer, which would lead to the peeling of perovskite films.^{3,4} Although the device is encapsulated, the temperature change (from -40 °C to 85 °C) still has a huge effect on the interface contact of the perovskite layer and charge transport layer of the encapsulated devices. Therefore, thermal cycling shows a faster degradation than those after damp heat testing.

5. In Figure 5, devices with Spiro were used for thermal stability tests. Since Spiro itself is not thermally stable, it is hard to use such a structure to compare the thermal stability of encapsulation materials.

Response: Thank you for your question. We are sorry for inadvertently omitting critical information. Actually, we prepare the normal devices with the spiro-MeOTAD hole-transporting layer to obtain higher efficiency in J - V measurement. When we conduct the damp heat stability test, the poly(triaryl amine) (PTAA) is used as the hole-transporting layer instead of spiro-MeOTAD in this manuscript.⁵⁻⁷ Therefore, the normal devices show good stability under the damp heat test.

In the revised manuscript, the relative discussion has been added on page 13 in the main text (also copied as below).

“The poly(triaryl amine) (PTAA) is used as the hole transporting layer instead of spiro-MeOTAD due to the good thermal stability.”

Reviewer #2:

This manuscript reports a new transparent self-crosslinked polymer (CFDP) as an encapsulant material for Perovskite Solar Cells. The material shows good hydrophobic properties and ability to prevent lead leakage. The stability result is impressive, passing both thermal cycling and damp heat. The author claims that the advantage is its good thermal management ability leading to effective heat dissipation under thermal stress, leading to lower level of perovskite composition. The author should include more in-depth discussion, including but not limited to ability for outgas suppression and mechanical properties enhancement. In addition, the comparison between CFDP and UV resin is not enough to showcase the advantages of CFDP. The author should compare with other reported successful encapsulation (which pass IEC tests, such as PIB/POE) from the literatures as well. Therefore, a revision is required.

Response: Thanks very much for the recognition. We are willing to make revisions based on the reviewer’s comments. The concerns mentioned here is similar to the questions below. Thus, the specific modifications will be discussed item by item later.

1. The only alternative encapsulant is UV resin and it is already known that UV resin can damage the device. Others have reported non-transparent polymer PIB or transparent polymer POE for encapsulating perovskite and tandem devices, all showing negligible device performance change (even at higher temperature) and passing the IEC test. Simply comparing between CFDP and UV Resin is not sufficient to showcase the advantages. **(a)** A more comprehensive analysis on adhesion, heat dissipation, stability with other polymer materials should be presented. Besides, **(b)** encapsulation process complexity and **(c)** cost should be compared as well.

Response: Thanks for the suggestion. The question can be divided into three parts.

Question (a): we have added the typical polymer encapsulant (POE) to further reveal the advantages of the CFDP polymer. We compare the adhesion, heat dissipation, and stability of UV resin, POE, and CFDP in **Figure R4-6** and **Table R1**. First, we observe the adhesion strength of CFDP (0.28 Mpa) is stronger than UV resin (0.19 Mpa). Although the adhesion strength of CFDP is weaker than POE (0.56 Mpa), the CFDP encapsulation avoids the thermal pressing process under vacuum environment. Second, the CFDP has the maximum thermal conductivity ($0.35 \text{ W m}^{-1} \text{ K}^{-1}$) compared with UV resin ($0.18 \text{ W m}^{-1} \text{ K}^{-1}$) and POE ($0.30 \text{ W m}^{-1} \text{ K}^{-1}$), which ensures its good heat dissipation capacity. Finally, the target encapsulated devices have excellent damp heat stability and thermal cycling stability compared with UV resin and POE.

Figure R4 The adhesion strength of UV resin, POE, and CFDP.

Figure R5 a, IR thermal images of encapsulated perovskite films during a cooling test. **b**, The surface temperature change of encapsulated perovskite films versus time.

Figure R6 Efficiency evolution of PSCs under (a) damp heat test at 85 °C and 85% relative humidity and (b) thermal cycling test from -40 °C and 85 °C. Error bars represent the statistical results of the three devices. The epoxy resin is used for edge sealing for better stability in all encapsulated methods.

Question (b) about the complexity of the encapsulation process: The hot pressing under the vacuum environment is required for POE encapsulation. The lamination at 120-150 °C for 15 min is the encapsulation process of POE. In addition, encapsulation with UV resin requires ultraviolet curing equipment. The encapsulated device is exposed to UV light for 5-10 min. Therefore, the encapsulation process with POE and UV resin is highly dependent on equipment. The target encapsulation is conducted at room temperature. The cross-linking process can be achieved at room temperature and does not require high temperature or UV light to initiate polymerization.

Question (c) about the cost analysis: The estimated material cost for the CFDP layer: Combining the estimated usage for ~60 μm thick polymer (~20 g TMPMA m⁻²; ~450 g F₃ m⁻²; ~130 g D₄ m⁻²; ~5 g TMAH m⁻²; ~10 g DBTDL m⁻²) and the quotation

of raw materials from industrial vendors with large amounts (\$5.53 for 1 kg TMPMA; \$2.76 for 1 kg F₃; \$4.15 for 1 kg D₄; \$1.80 for 1 kg TMAH; \$1.38 for 1 kg DBTDL. The data is obtained from Alibaba company). The cost could be calculated to be ~1.91 \$ m⁻², which is more expensive than commercialized POE (~1.70 \$ m⁻²) films and cheaper than the price of UV resin (~23.87 \$ m⁻²).^{8,9} The unit price might be reduced if the CFDP could be industrialized. In addition, encapsulation with POE requires hot pressing equipment, and encapsulation with UV resin requires UV curing equipment, which will also increase the cost.

Table R1. Performance, encapsulation process, and cost summary of perovskite encapsulant.

Type	Adhesion strength (Mpa)	Thermal conductivity (W m ⁻¹ K ⁻¹)	Stability (damp heat test and thermal cycling test)	Process complexity	Cost (\$ m ⁻²)
UV resin	0.19	0.18	80%/75%	UV light	23.87
POE	0.56	0.30	95%/93%	Vacuum hot pressing	1.70
CFDP	0.28	0.35	98%/95%	Simple	1.91

In the revised version, **Figure R4** is added as **Supplementary Figure 11**; **Figure R6** is added as **Supplementary Figure 25**; the encapsulation process complexity and cost analysis have been added to **Supplementary Note 1** in supplementary information and the relative discussion has been added on page 5 in the main text (also copied as below).

“Finally, the CFDP polymer gel had a simple encapsulation process and low cost compared with other encapsulants (Supplementary Note 1).”

2. No details are provided for the deposition of FDP and CFDP layers on glass substrates.
- (a) Does it involve application of FDP, adding DBTDL, and waiting for solidification? (b) Does it have a similar solidification mechanism of a 2-part epoxy? (c) Is there any outgas from the solidification process from CFDP? (d) The curing of CFDP is not

provided in the paper either, such as how long for CFDP to become a solid and what curing conditions are required. These are useful for understanding the encapsulation adhesion mechanisms.

Response: Thanks for your question. The question can be divided into four parts.

Question (a): The deposition process involves the application of FDP, adding DBTDL, and waiting for solidification. First, the FDP polymer (1 g) and DBTDL (0.01g, 1wt%) are mixed in a glass bottle. After stirring for 1 min and waiting for 3-5 min, the FDP and CFDP are spin-coated at 2000 rpm for 20 s on the glass substrates. After 30 min, the CFDP is completely cured. Finally, the FDP and CFDP with a certain thickness deposit on the glass substrates successfully.

Question (b): The solidification mechanism of CFDP is similar to the catalytic curing of 2-part epoxy. The solidification mechanism of the 2-part epoxy mainly includes two types: reactive curing and catalytic curing. Reactive curing is mainly through the reaction of "active hydrogen" with epoxy groups, and the resin is cured and crosslinked. Catalytic curing is mainly to open the epoxy group to form a homopolymer with ether bonds as the main structure. The catalyst only plays a catalytic role and does not participate in the cross-linking itself.

Question (c): There are not any outgas from the solidification process from CFDP. It is worth mentioning that, the hot-pressing encapsulation process generates hot outgas and the UV-curing encapsulation process generates harmful outgassed vapor.¹⁰ Thus, the CFDP encapsulation reflects the progressiveness from the perspective of waste gas generation.

Question (d): There are three stages in the solidification process, including the Liquid-operating time, Gel-entering curing, and Solids-final curing.¹¹ The CFDP becomes solid in 30-40 min after adding the catalyst DBTDL into the polymer system. The curing process can be completed at room temperature.

In the revised version, the relative discussion has been added on page 4 in the supplementary information (also copied as below).

“The encapsulation process involved the application of FDP, adding DBTDL, and

waiting for solidification. First, the FDP polymer (1 g) and DBTDL (0.01g, 1wt%) were mixed in a glass bottle. After stirring for 1 min and waiting 3-5 min, the CFDP was spin-coated at 2000 rpm for 30 s on the glass substrates. After 30 min, the CFDP was completely cured. The encapsulation process was completed before the CFDP was fully cured. There were not any outgas from the solidification process from CFDP.”

3. What are the thickness of CFDP and FDP layers and how they are compared to other encapsulates such as UV resin?

Response: Thanks for your question. As shown in **Figure R7**, the thickness of CFDP layers is $\sim 60.25 \mu\text{m}$ and the thickness of UV resin is slightly thicker than the CFDP, which is $\sim 65.67 \mu\text{m}$. Furthermore, it is difficult to measure the thickness of FDP due to the high viscosity and liquid properties.

Figure R7. The film thickness of UV resin and CFDP polymer obtained by DektakXT stylus profiler.

In the revised version, the relative discussion has been added on page 4 in the supplementary information (also copied as below).

“The thickness of CFDP layer was $\sim 60.25 \mu\text{m}$ and the thickness of UV resin layer was slightly thicker than the CFDP, which was $\sim 65.67 \mu\text{m}$.”

4. CFDP shows enhanced hydrophobicity and adhesion strength. **(a)** As CFDP is formed by incorporating DBTDL into FDP, please explain how DBTDL improves the hydrophobicity and adhesion strength. **(b)** Also, readers are more interested in how CFDP is compared with UV resin and other encapsulants regarding those two properties. FDP is not used as an encapsulant in this paper.

Response: Thanks for your question. The question can be divided into two parts.

Question (a): The reason for the enhanced hydrophobicity is the different permeability of the droplets on the surface of FDP and CFDP polymers, which is verified in the previous study.¹² Before incorporating DBTDL into FDP, the FDP is a viscous polymer with some degree of fluidity. When water droplets arrive at the surface of polymer, water droplets can easily invade. After adding DBTDL, the CFDP turns into a solid and suppresses water invasion efficiently. Therefore, the contact angle of CFDP is bigger than that of FDP.

The reason for the enhanced adhesion strength is the difference in the molecular weight and structure of FDP and CFDP. Before incorporating the catalyst DBTDL into FDP, FDP polymer has a low molecular weight with a linear structure, so the bonding strength to the substrate is poor relatively. After incorporating DBTDL, the FDP polymer is crosslinked. The molecular weight of CFDP is enhanced, showing a network structure. Due to the entanglements of polymer molecules and the increased molecular weight, the adhesion strength of CFDP is significantly improved.^{13,14}

Question (b): To further compare the hydrophobicity and adhesion strength between the CFDP encapsulants, UV resin and POE encapsulants, we perform the contact angle tests and single-lap tensile shear tests. Firstly, we observe the CFDP shows the biggest contact angle (121.29°) due to the low surface energy compared with the UV resin (66.92°) and POE (85.26°) in **Figure R8-R10**. In addition, the adhesion strength is calculated via the following formula: $\tau = F/A$, where τ is the adhesion strength, F is the maximum load, and A is the bonding area. The adhesion strength of UV resin and POE is 0.19 Mpa and 0.56 Mpa in **Figure R11**, while the adhesion strength of CFDP is 0.28 Mpa. Therefore, the CFDP shows excellent hydrophobicity and strong adhesion strength compared with the UV resin and POE.

Figure R8 The water contact angles of UV resin film as a function of time.

Figure R9 The water contact angles of POE film as a function of time.

Figure R10 The water contact angles of CFDP film as a function of time.

Figure R11 The adhesion strength of UV resin, POE, and CFDP.

In the revised version, **Figure R8** is added as **Supplementary Figure 9**; **Figure**

R9 is added as **Supplementary Figure 10**; **Figure R11** is added as **Supplementary Figure 11** in the supplementary information and the relative discussion has been added on page 5 in the main text (also copied as below).

“ In addition, CFDP showed excellent hydrophobicity and strong adhesion strength compared with other encapsulants (Supplementary Figs. 9-11).”

5. In Table S1, the author listed the water vapor transmission rate of CFDP. How is this value derived, from reference or measurement?

Response: Thanks for your question. The value of CFDP derives from the measurement via the Water Vapor Transmission Rate Test System (C306H). At a selected temperature (25 °C) and humidity (90% RH), according to the standard of ASTM F1249, the polymer film (the area is 25 cm² and thickness is 1 mm) is sealed between a wet chamber and a dry chamber. An infrared moisture sensor measures the moisture transmitted through the material tested.

In the revised manuscript, the relative discussion has been added on page 4 of the supplementary information, also copied as below.

“The water vapor transmission rate of CFDP film (the area was 25 cm² and thickness was 1 mm) was derived from the measurement via the Water Vapor Transmission Rate Test System (C306H) at a selected temperature (25 °C) and humidity (90% RH).”

6. Please include CFDP-only results from Figure S11-12 in Figure 2. This can give a more straight forward comparison to show the advantage of CFDP over UV resin. Again please compare with other encapsulants.

Response: Thanks for the suggestion. We have moved the CFDP-only results in **Figure R12** to **Figure 2** for the straight comparison in the revised version. Furthermore, we have compared the encapsulant such as POE to further reveal the advantage of the target encapsulation strategy. We observe that the CFDP encapsulation and target encapsulation have a better heat dissipation capacity compared with UV resin encapsulation and POE encapsulation.

Figure R12 a, IR thermal images of encapsulated perovskite films during a cooling test. **b**, The surface temperature change of encapsulated perovskite films versus time.

In the revised manuscript, **Figure R12** is added as **Figure 2a-b** and **Supplementary Figure 14** and the relative discussion has been added on page 6 in the main text (also copied as below).

“In addition, the POE encapsulation was also compared (Supplementary Fig. 14). The CFDP encapsulation and target encapsulation had a better heat dissipation capacity compared with UV resin encapsulation and POE encapsulation.”

7. In Figure 2a and 2b, **(a)** why target encapsulation has a faster cooling rate at the beginning and a slower cooling rate at the end while UV resin cooling rate is quite constant. **(b)** Why the edge of both samples has a lower temperature even from the beginning?

Response: Thanks for your question. The question can be divided into two parts.

Question (a): The results may be due to the imprecision of a single experiment, which will cause some misunderstandings. To ensure the reliability of the experiment, we test each sample three times and calculate the average value of temperature and add the error bars. Besides, we have extended the testing time. As shown in **Figure R13**, regardless of UV encapsulation, CFDP encapsulation, or target encapsulation, the cooling trend is basically similar, showing a faster cooling rate at the beginning and a slower cooling rate at the end. The phenomenon is similar to the previous report.¹⁵ Due to larger thermal conductivity, the inflection point of temperature trend in target encapsulation is earlier than the encapsulation with CFDP and UV resin.

Question (b): Since the perovskite film is encapsulated, the center temperature is the highest and the heat in the center is difficult to transfer out due to the cover glass. Heat transfers faster at the edges than at the center, so the edge of both samples has a lower temperature even from the beginning. Infrared thermal imaging photos in the literature also prove our point.^{16,17}

Figure R13 a, IR thermal images of encapsulated perovskite films during a cooling test. **b,** The surface temperature change of encapsulated perovskite films versus time.

8. In Figure 2f-h, both SEM and XRD show that after 85 °C thermal stress for 500 hour, UV resin encapsulated sample shows more severe perovskite decomposition due to poor heat dissipation ability. However, when the sample is under constant thermal stress, the heat exchange should be different to the previous cooling conditions. Is it possible to directly use the infrared camera to measure the temperature of the samples during thermal stressing. In this way, it may be possible to directly check whether the temperature of UV resin encapsulated sample is much higher all the time, being a more direct evidence to show heat accumulation.

Response: Thanks for the suggestion. To reveal the heat dissipation capacity of encapsulated perovskite films under thermal equilibrium conditions, the infrared camera is used to monitor the temperature of the samples during continuous heating at 85 °C in **Figure R14**. After 10 min, the temperature of the UV resin encapsulated sample is higher than that of the target encapsulated sample obviously. The average temperature is 83.1 °C and 81.2 °C for the UV resin and target encapsulated sample, respectively, which shows more heat accumulation for the UV resin encapsulated sample. Therefore, the target encapsulated sample has a lower temperature and better heat dissipation capability than the UV resin encapsulated sample under thermal equilibrium conditions.

Figure R14 (a) IR thermal images of UV resin and target encapsulated perovskite films under thermal stress. (b) Statistical histogram of the average temperature. The encapsulated sample reaches the thermal equilibrium state when the average temperature reaches the stable value.

In the revised manuscript, **Figure R14** is added as **Supplementary Figure 15** and the relative discussion has been added on page 7 in the main text (also copied as below).

*“Furthermore, we used the infrared camera to monitor the temperature of the encapsulated samples during continuous heating at 85 °C in **Supplementary Fig. 15**. After 10 min, the UV resin encapsulated sample still maintained a higher average temperature (83.1 °C) than the target encapsulation (81.2 °C) under thermal equilibrium conditions.”*

9. Do you think the more significant degradation also comes from weaker outgas suppression ability of UV resin? Please discuss on this considering material density/thickness, adhesion, outgas permeability and etc.

Response: Thanks for your question. The outgassed vapor of UV resin does damage the perovskite film during the UV-curing process, thus affecting the stability of encapsulated devices.⁹ We compare the material density/thickness, adhesion, and outgas permeability in **Table R2**. Compared with the CFDP, the UV resin has a thicker thickness, weaker adhesion strength, and weaker outgas suppression ability. Therefore, damage of the outgassed vapor to perovskite films during encapsulation with UV resin is unavoidable. Compared to the effect of heat dissipation on device stability, the outgas is a short-term effect, which only generates during the encapsulation process due to UV irradiation. However, the heat gradually accumulates as the encapsulated device works,

which has a continuous impact on the stability of device. Therefore, the heat dissipation capacity of encapsulant is more important than the outgas suppression ability for the stability of encapsulated devices.

Table R2. Performance summary of perovskite encapsulant.

Type	Density (g/mL)	Thickness (μm)	Adhesion (Mpa)	Outgas suppression
UV resin	1.16	~ 65.67	0.19	Weak
CFDP	1.25	~ 60.25	0.28	Strong

10. On page 8 and in Figure 4, Pb leakage measurement, it is clearly that the breaking of all three samples are different. **(a)** Is it because the metal ball drops differently or does it mean CFDP has a stronger mechanical property? or is CFDP thicker than UV resin so it can provide a good protection mechanically? The author has mentioned enhanced mechanical strength due to the reaction of -OH in the glass and -Si-O-CH₃ in CFDP. However, what I understand is this reaction can improve the adhesion between the encapsulant and the glass but why the whole device becomes tougher upon damage? **(b)** Also please compare the mechanical strength of other encapsulants vs CFDP as well.

Response: Thanks for your question. The question can be divided into two parts.

Question (a): To ensure the consistency of the experiment, the metal ball drops from a constant height. In addition, by measuring the thickness of the film, the thickness of the CFDP encapsulation layer ($\sim 60.25 \mu\text{m}$) is slightly thinner than the UV resin layer ($\sim 65.67 \mu\text{m}$). We analyze the reason that the whole device becomes tougher upon damage. From the perspective of polymer structure, UV resin encapsulant is an unsaturated resin containing a large amount of C=C groups. After the material is fully cured, the hardness is very high. Therefore, the material does not have the effect of buffering. As a comparison, CFDP is a polymer gel, containing a large number of Si-O groups. The Si-O bond has a small internal rotation barrier, which endows the good flexibility and elasticity of polymer.¹⁸ When the metal ball hits the glass, the flexible encapsulation materials have good buffering effect, thereby reducing the damage to the underlying glass by impact. From the perspective of chemical reaction, the reaction of -OH in the glass and -Si-O-CH₃ in CFDP improve the adhesion between the encapsulant

and the glass. More importantly, the reaction can achieve interfacial toughening and enhance the mechanical reliability of cover glass.¹⁹ Therefore, by using polymer gel CFDP, a "molecular bridge" can be built between the interface of inorganic (glass) and organic substances (polymer gel) to connect the two materials to increase the adhesive strength, achieving a better buffer effect and improving the impact resistance of cover glass.

Question (b): We further compare the mechanical strength of other encapsulants (POE) via the hail experiment in **Figure R15**. We observe that the cover glass on the metal electrode side is seriously damaged, while the glass on the ITO side remains relatively intact. Although POE also provides good protection for the cover glass, the efficient suppression of lead leakage cannot be achieved due to the absence of functional groups to chelate lead.

Figure R15 The typical process for the preparation of damaged PSCs is based on the device with POE encapsulation.

In the revised manuscript, **Figure R15** is added as **Supplementary Figure 31** and the relative discussion has been added on page 10 in the main text (also copied as below).
“First, the polymer gel chemically anchors to the glass substrate due to the reaction between the -OH groups on the substrate and the -Si-O-CH₃ groups in the CFDP, which improves the adhesion between the encapsulant and the glass. More importantly, the reaction achieves interfacial toughening and enhances the mechanical reliability of the cover glass. As a result, by using polymer gel with good flexibility and elasticity, a "molecular bridge" can be built between the interface to increase the adhesive strength, achieving a better buffer effect and improving the impact resistance of cover glass.”

*“In addition, compared with the POE encapsulation (**Supplementary Figure 31**), the CFDP encapsulant achieved the better protection for the encapsulated PSCs.”*

11. Please state whether the encapsulated perovskite film in this studies have cover glass or not for SEM, XRD, PL, temperature cooling test and lead leakage test. If there is a cover glass, how did you remove the encapsulants and cover glass after stressing to measure SEM, XRD and PL? Will removing the encapsulation damage the film as well?

Response: Thank you for your question. The encapsulated perovskite films in this study have cover glass for SEM, XRD, PL, temperature cooling test, and lead leakage test. For PL testing, the encapsulants and cover glass do not need to be removed. We can test from the substrate glass with a laser, which is consistent with the previous report.²⁰ For SEM and XRD, the encapsulation layer must be removed to reveal the effect of encapsulation on perovskite films. In the process of removing encapsulation, it is indeed easy to damage the perovskite films. To ensure the accuracy of the experiment, we tried many strategies and selected a method that caused relatively little damage to the perovskite films. To better remove the cover glass and encapsulation layer, we use half of the glass to cover the perovskite films instead of the previous full coverage.²¹ We further chisel encapsulated glass and encapsulant with a scalpel carefully in **Figure R16a-b**.²² We observe that there are a small number of holes on the surface of UV resin and target encapsulated perovskite films (**Figure R16c-d**). In the manuscript, we selected the part of the film without obvious damage for comparison to better reveal the effect of heat dissipation capability on the encapsulated perovskite films, while excluding the influence of removing the encapsulation on the perovskite films.

Figure R16 (a) Schematic diagram of removing the encapsulation for preparing the SEM samples. (b) Photographs of the encapsulated perovskite films before and after the encapsulation removal. Top-view SEM images of perovskite films after (c) UV resin encapsulation and (d) target encapsulation removal. The pink arrow points to the holes.

In the revised manuscript, **Figure R16** is added as **Supplementary Figure 17** and the relative discussion has been added on page 7 in the main text (also copied as below).

“We carefully removed the cover glass and encapsulation layer and selected the part of perovskite films without obvious damage for comparison (Supplementary Fig. 17).”

12. In Figure 5, the second structure is based on MA and Spiro, which is more vulnerable to thermal stress. Why is the stability trend similar to the first inorganic CsPbI₂Br based devices? The thermal stabilities of MA and Cs based perovskite as well as NiO_x and Spiro are clearly different even when the same encapsulation method is used.

Response: Thanks for your question. we are sorry for inadvertently omitting critical

information. Actually, we prepare the normal devices with the spiro-MeOTAD hole-transporting layer to obtain higher efficiency in J - V measurement. When we conduct the damp heat stability test, the poly(triaryl amine) (PTAA) is used as the hole-transporting layer instead of spiro-MeOTAD in this manuscript.^{5,6} Furthermore, the 1-alkyl-4-amino-1,2,4-triazolium as the additive was added into the precursor solution and improved the quality of perovskite films, which is proved by the previous research.²³ The incorporation of additives have mentioned in the experimental section in the previous version of the manuscript. Therefore, the normal devices based on the MAPbI₃ perovskite show good stability under the damp heat test.

Even so, the stability trend of the MAPbI₃ devices is not similar to the inorganic CsPbI₂Br based devices. The CsPbI₂Br devices with target encapsulation showed slower degradation and retained 91% of the initial PCE, while the MAPbI₃ devices with target encapsulation maintained 83% of their initial PCE. Whether it is UV encapsulation or target encapsulation, the MAPbI₃ devices show a faster decay than the CsPbI₂Br devices.

In the revised manuscript, the relative discussion has been added on page 13 in the main text (also copied as below).

“The poly(triaryl amine) (PTAA) is used as the hole-transporting layer instead of spiro-MeOTAD due to the good thermal stability.”

Reviewer #3:

This manuscript introduced an innovated perovskite encapsulation materials by synthesis CFDP and its composite blends, which enriches the choice for the encapsulants of perovskite solar cells. The authors systematically characterize the validity of the synthetic procedure of CFDP and its intrinsic physical properties. The CFDP composite encapsulated device maintained 98% and 95% of its initial PCE (21.91% after CFDP encapsulation) after 1000h damp heat test and 220 cycles of cyclic thermal shocks respectively, which satisfy the IEC61215:2016 standard. I think the work is interesting and important for the PVSK community, and hence it can be

considered for publication on Nature Communication.

Response: We thank the reviewer for the encouraging comments. We are willing to make revisions based on the reviewer's comments and suggestions.

Some questions need to be addressed:

1. The authors describe the CFDP as a thermally stable molecule. However, the CFDP sample starts to lose its weight when the temperature only raised to 150 °C while 10% of its initial weight was lost at 505.17 °C, according to the TGA characterization in Fig.1(g). A further explanation is needed about why the crosslinked molecule shows a lower initial degradation temperature with its potential impact.

Response: Thank you for your question. I speculate that the small molecule silane monomers TMPMA may be present in CFDP due to lack of purification for CFDP. Therefore, the crosslinked polymer CFDP starts to lose its weight at 150 °C due to the decomposition of the silane monomers TMPMA in the system.²⁴ To prove our conjecture, CFDP is further purified to remove the TMPMA. We retest the thermal decomposition temperature of CFDP (the temperature at 5% mass loss) in **Figure R17**. The thermal decomposition temperature increases from 301.11 °C (FDP) to 484.28 °C (CFDP), indicating the excellent thermal stability of the CFDP. Our encapsulation strategy is performed at room temperature and does not require the process of high temperature and vacuum environment, so the degradation does not occur during the encapsulation process.

Figure R17 Thermogravimetric analysis (TGA) curves of FDP and CFDP.

In the revised manuscript, **Figure R17** is added as **Figure 1g** and the relative discussion has been added on page 5 in the main text (also copied as below).

“The thermal decomposition temperature (the temperature at 5% mass loss) increased from 301.11 °C (FDP) to 484.28 °C (CFDP), indicating the excellent thermal stability of the CFDP.”

2. According to the description in line 117, BN was incorporated in the CDFP gel to improve the thermal conductivity. However, the UV resin did not add equivalent BN before UV cured, which makes the optimized thermal dissipation less convective when the data of the intrinsic thermal conductivity of CFDP cannot be found in the manuscript.

Response: Thank you for your suggestion. Actually, the CFDP-only results have already been provided in **Figure S11-12** of supplementary information previously. The results are also shown in **Figure R18**. To better reveal the advantage of CFDP over UV resin, we move the CFDP-only results to the main text. We observe that the heat dissipation capacity of CFDP encapsulation is stronger than UV resin encapsulation via infrared thermal images. Furthermore, the intrinsic thermal conductivity of CFDP is higher than the UV resin at a given fixed temperature (Figure R19). Therefore, the encapsulated devices with CFDP encapsulation show a better heat dissipation effect than the devices with UV encapsulation even without adding BN. To further improve the heat dissipation effect, we use the composite gel to encapsulate the PSCs, revealing the significance of optimized thermal management for the encapsulated PSCs.

Figure R18 a, IR thermal images of encapsulated perovskite films during a cooling test. **b**, The surface temperature change of encapsulated perovskite films versus time.

In the revised manuscript, **Figure R18** is added as **Figure 2a-b**. Furthermore, we supplement the data of the intrinsic thermal conductivity of CFDP in **Figure R19** to reveal the optimized thermal dissipation.

Figure R19 Thermal conductivities of polymer sealant under different temperatures.

In the revised manuscript, **Figure R19** is added as **Figure 2c** and the relative discussion has been added on page 7 in the main text (also copied as below).

“At room temperature, the thermal conductivities for UV resin, CFDP, and CFDP composite were $0.15 \text{ W m}^{-1} \text{ K}^{-1}$, $0.35 \text{ W m}^{-1} \text{ K}^{-1}$, and $0.51 \text{ W m}^{-1} \text{ K}^{-1}$, respectively. When the temperature reached $55 \text{ }^\circ\text{C}$, the thermal conductivities reduced to $0.12 \text{ W m}^{-1} \text{ K}^{-1}$ (UV resin), $0.29 \text{ W m}^{-1} \text{ K}^{-1}$ (CFDP), and $0.49 \text{ W m}^{-1} \text{ K}^{-1}$ (CFDP composite). At $85 \text{ }^\circ\text{C}$, the thermal conductivities further decreased to $0.11 \text{ W m}^{-1} \text{ K}^{-1}$ (UV resin), $0.26 \text{ W m}^{-1} \text{ K}^{-1}$ (CFDP), and $0.45 \text{ W m}^{-1} \text{ K}^{-1}$ (CFDP composite).”

3. In Line 199, the author mentioned that the C=O bond in the CFDP molecule could establish strong interaction with the Pb^{2+} in perovskite, (a) which may not correspond well with Fig2. (g). The SEM image above did not show drastic changes when the CFDP was removed from the surface of perovskite. (b) I think the authors should describe the preparation process of the SEM sample, for viscous encapsulant are usually hard to remove when the encapsulation is done. (c) Meanwhile, is it possible that the strong chemical bond formed between Pb^{2+} and C=O could reduce the concentration of PbI_2 on the surface of perovskite layer (PbI_2 removed alone with the removal of CFDP encapsulant)?

Response: Thanks for your question. The question can be divided into three parts.

Question (a): The encapsulation process of perovskite film is completely different from the process of lead leakage. The encapsulation process is similar to the post-modification of perovskite films. At that time, the perovskite film is solid, and the

encapsulation process does not affect the crystallization and morphology of perovskite films.²⁵ Thus, the SEM image of perovskite films will not change drastically. For the lead leakage process, the device is invaded by water and lead is dissolved. The C=O bond in the CFDP molecule can strongly interact with the Pb²⁺ dissolved in water. Therefore, the two processes are not comparable.

Question (b) about the preparation process of the SEM sample: In the process of removing encapsulation, it is easy to damage the perovskite film. To ensure the accuracy of the experiment, we tried many strategies and selected a method that caused relatively little damage to the perovskite films. To better remove the cover glass and encapsulation layer, we use half of the glass to cover the perovskite films instead of the previous full coverage.²¹ We further chisel encapsulated glass and encapsulant with a scalpel carefully in **Figure R20a-b**.²² As shown in **Figure R20c-d**, we observe that there are a small number of holes on the surface of UV resin and target encapsulated perovskite films. In the manuscript, we selected the part of the film without obvious damage for comparison to better reveal the effect of heat dissipation capability on the encapsulated perovskite films, while excluding the influence of removing the encapsulation on the perovskite film.

Question (c): I agreed the reviewer's comment that the process of removing the encapsulation layer may reduce the concentration of PbI₂ on the surface of the perovskite layer. However, we do not observe significantly the perovskite and PbI₂ on the encapsulated polymer after removing the encapsulation in **Figure R20b**. Furthermore, the morphology of perovskite films with target encapsulation does not change much compared to that with UV resin encapsulation in the manuscript (**Fig. f-g**), revealing less decomposition of perovskite with target encapsulation. In the revised version, to present our views more rigorously, we have re-written our conclusion: "Excellent thermal management is an important reason for reducing the concentration of PbI₂ on the perovskite surface."

Figure R20 (a) Schematic diagram of removing the encapsulation for preparing the SEM samples. (b) Photographs of the encapsulated perovskite films before and after the encapsulation removal. Top-view SEM images of perovskite films after (c) UV resin encapsulation and (d) target encapsulation removal. The pink arrow points to the holes.

In the revised manuscript, **Figure R20** is added as **Supplementary Figure 17** and the relative discussion has been added on page 7 and page 8 in the main text (also copied as below).

“We carefully removed the cover glass and encapsulation layer and selected the part of perovskite films without obvious damage for comparison (Supplementary Fig. 17).”

“The cross-linked polymer network loaded with a thermally conductive filler effectively promotes heat transfer and mitigates the potential impact of heat accumulation, which is an important reason for reducing the concentration of PbI_2 on the perovskite surface.”

4. The encapsulation process with a lamination at 120 °C for 15 min does not show any

damage to the performance of the solar cell device in terms of PCE. This is usually difficult for the PVSK device. Does this have something to do with the CFDP itself or the BN added into the encapsulant?

Response: Thanks for your question. The reviewer may have some misunderstandings about our encapsulation strategy. The lamination at 120 °C for 15 min is the encapsulation process of POE encapsulation, which would damage the performance of the PSCs partly. In a vacuum environment, the perovskite undergoes lattice distortion, which affects the efficiency of encapsulated devices.¹ For our encapsulation method, the CFDP achieves the cross-linked at room temperature. There are no damaging external conditions such as high temperature, vacuum environment, or UV radiation during the encapsulation process. The target encapsulation strategy achieves non-destructive encapsulation at room temperature. Therefore, non-destructive encapsulation is about the CFDP itself. In addition, the purpose of introducing BN into CFDP is to improve the thermal conductivity of encapsulant, optimize the thermal management of packaged devices, and further improve the stability of encapsulated PSCs.

References

- 1 Guo, R. et al. Degradation mechanisms of perovskite solar cells under vacuum and one atmosphere of nitrogen. *Nat. Energy* **6**, 977-986, (2021).
- 2 Shi, L. et al. Gas chromatography–mass spectrometry analyses of encapsulated stable perovskite solar cells. *Science* **368**, eaba2412, (2020).
- 3 Mei, A. et al. Stabilizing perovskite solar cells to IEC61215:2016 standards with over 9,000-h operational tracking. *Joule* **4**, 2646-2660, (2020).
- 4 Zhao, J. et al. Strained hybrid perovskite thin films and their impact on the intrinsic stability of perovskite solar cells. *Sci. Adv.* **3**, eaa05616.
- 5 Wang, M. et al. Rational selection of the polymeric structure for interface engineering of perovskite solar cells. *Joule* **6**, 1032-1048, (2022).
- 6 Zhao, Y. et al. Inactive (PbI₂)₂RbCl stabilizes perovskite films for efficient solar cells. *Science* **377**, 531-534, (2022).
- 7 Zai, H. et al. Sandwiched electrode buffer for efficient and stable perovskite solar cells with dual back surface fields. *Joule* **5**, 2148-2163, (2021).
- 8 Xiao, X. et al. Lead-adsorbing ionogel-based encapsulation for impact-resistant, stable, and lead-safe perovskite modules. *Sci. Adv.* **7**, eabi8249, (2021).
- 9 Ma, S. et al. 1000 h operational lifetime perovskite solar cells by ambient melting encapsulation. *Adv. Energy Mater.* **10**, 1902472, (2020).
- 10 Dong, Q. et al. Encapsulation of Perovskite Solar Cells for High Humidity Conditions. *ChemSusChem* **9**, 2597-2603, (2016).
- 11 Chemtob, A., Versace, D.-L., Belon, C., Croutxé-Barghorn, C. & Rigolet, S. Concomitant Organic-Inorganic UV-Curing Catalyzed by Photoacids. *Macromolecules* **41**, 7390-7398, (2008).
- 12 Krainer, S. & Hirn, U. Contact angle measurement on porous substrates: Effect of liquid absorption and drop size. *Colloids and Surfaces A: Physicochemical and Engineering Aspects* **619**, 126503, (2021).
- 13 Kajtna, J., Golob, J. & Krajnc, M. The effect of polymer molecular weight and crosslinking reactions on the adhesion properties of microsphere water-based acrylic pressure-sensitive adhesives. *Int. J. Adhes. Adhes.* **29**, 186-194, (2009).
- 14 Yu, J., Cheng, B. & Ejima, H. Effect of molecular weight and polymer composition on gallol-functionalized underwater adhesive. *Journal of Materials Chemistry B* **8**, 6798-6801, (2020).
- 15 Li, M.-D. et al. Thermal management of chips by a device prototype using synergistic effects of 3-D heat-conductive network and electrocaloric refrigeration. *Nat. Commun.* **13**, 5849, (2022).
- 16 Choi, K. et al. Heat dissipation effects on the stability of planar perovskite solar cells. *Energy Environ. Sci.* **13**, 5059-5067, (2020).
- 17 Pei, F. et al. Thermal management enables more efficient and stable perovskite solar cells. *ACS Energy Lett.* **6**, 3029-3036, (2021).
- 18 Hao, S.-M. et al. Lithium-conducting branched polymers: New paradigm of solid-state electrolytes for batteries. *Nano Lett.* **21**, 7435-7447, (2021).
- 19 Dai, Z. et al. Interfacial toughening with self-assembled monolayers enhances perovskite solar cell reliability. *Science* **372**, 618, (2021).
- 20 Lin, Z. Q. et al. Mediating the local oxygen-bridge interactions of oxysalt/perovskite interface

- for defect passivation of perovskite photovoltaics. *Nano-Micro Letters* **13**, 177, (2021).
- 21 Bai, Y. et al. Oligomeric silica-wrapped perovskites enable synchronous defect passivation and grain stabilization for efficient and stable perovskite photovoltaics. *ACS Energy Lett.* **4**, 1231-1240, (2019).
- 22 Chen, S. et al. Stabilizing perovskite-substrate interfaces for high-performance perovskite modules. *Science* **373**, 902, (2021).
- 23 Wang, S. et al. Water-soluble triazolium ionic-liquid-induced surface self-assembly to enhance the stability and efficiency of perovskite solar cells. *Adv. Funct. Mater.* **29**, 1900417, (2019).
- 24 Yamazaki, R., Karyu, N., Noda, M., Fujii, S. & Nakamura, Y. Quantitative measurement of physisorbed silane on a silica particle surface treated with silane coupling agents by thermogravimetric analysis. *J. Appl. Polym. Sci.* **133**, (2016).
- 25 Li, Z. et al. Organometallic-functionalized interfaces for highly efficient inverted perovskite solar cells. *Science* **376**, 416-420, (2022).

REVIEWER COMMENTS

Reviewer #1 (Remarks to the Author):

The authors have sufficiently addressed all of my comments.

Reviewer #2 (Remarks to the Author):

Thanks to the authors who have addressed most of the comments except the following

Response 2.2 and 2.3:

Could you please expand on how to control the encapsulant thickness? What is the optimum thickness? The thicker the better?

Please provide evidence for this statement, "There are not any outgas from the solidification process from CFDP"? The author explained that the curing mechanism is still related to the reaction of "active hydrogen" with epoxy groups. So what is the side product and the reaction then? Please provide details

For Table R1, it would be better to include another column of "overall processing time". Please discuss that CFDP encapsulation process actually takes longer time.

Response 2.9: any methods to quantify the outgas?

Response 2.12: It is confusing that Figure 5's JV curve shows the

spiro-based device (5c) while stability result is based on PTAA-based device (5d). Please replace the JV in Figure 5c with PTAA-based device result.

Reviewer #3 (Remarks to the Author):

The authors have addressed my previous concerns. The current version can be accepted.

Point-by-point Response to the Reviewers' Comments

Reviewer #1

The authors have sufficiently addressed all of my comments.

Response: We deeply appreciate the reviewing efforts and positive comments.

Reviewer #2:

Thanks to the authors who have addressed most of the comments except the following.

Response: Thanks very much for the recognition. We are willing to make revisions based on the reviewer's comments.

1. **(a)** Could you please expand on how to control the encapsulant thickness? **(b)** What is the optimum thickness? The thicker the better?

Response: Thanks for your question. The question can be divided into two parts.

Question (a): We control the thickness of the encapsulant via spin coating. Specifically, we apply the uncured encapsulant to the glass at different spin coating speeds (1000 rpm, 1500 rpm, 2000 rpm, 2500 rpm, and 3000 rpm). After the encapsulant is fully cured, the thickness of the encapsulant is measured by DektakXT stylus profiler (**Table R1**). In the process of the device encapsulation, we cover the perovskite solar cell with polymer-coated glass after spin coating. Then wait for the encapsulant to completely cure.

Table R1. The corresponding relationship between spin coating speed and encapsulant thickness.

Speed (rpm)	1000	1500	2000	2500	3000
Thickness (μm)	32.87	45.01	60.25	68.67	78.32

Question (b): The optimum thickness is $\sim 60.25 \mu\text{m}$. The thickness of the encapsulant is not as thick as possible. We optimize the thickness by testing the stability of encapsulated devices in **Figure R1**. When the encapsulant is too thin, the encapsulated function is not enough. When the thickness of encapsulant exceeds 60.25

μm , the stability of device changes little. However, the encapsulant will spill out as the cover glass is squeezed in the process of encapsulation if the encapsulant is too thick. It will cause serious material waste. Therefore, we set the optimum thickness of encapsulant as $\sim 60.25 \mu\text{m}$.

Figure R1 Relationship between the thickness of encapsulant and the stability of encapsulated devices. The encapsulated devices are aged at maximum power point under AM 1.5 illumination at $55 \pm 5 \text{ }^\circ\text{C}$ in the air for 100 h. The ordinate represents the normalized efficiency of the encapsulated devices after aging compared to the initial efficiency.

In the revised version, **Figure R1** is added as **Supplementary Figure 19b**; the relative discussion about question (a) has been added on page 3 in the supplementary information; and the relative discussion about question (b) has been added on page 8 in the main text (also copied as below).

“We controlled the thickness of the encapsulant via the spin coating and obtained the optimum thickness by adjusting the spin coating speed (2000 rpm).” (Page 3 in the supplementary information)

“The optimum thickness of CFDP encapsulation layer is $\sim 60.25 \mu\text{m}$ (Supplementary Fig. 19b).” (Page 8 in the main text)

2. Please provide evidence for this statement, “There are not any outgas from the solidification process from CFDP”? The author explained that the curing mechanism is still related to the reaction of “active hydrogen” with epoxy groups. So what is the side product and the reaction then? Please provide details

Response: Thanks for your insightful question. We apologize for some misunderstanding on this question. The encapsulant undergoes a condensation reaction under the action of the catalyst (DBTDL). Small amounts of side product (alcohol) are produced during the curing process. Therefore, CFDP may produce a small amount of

outgas according to the reaction mechanism. Fourier-transform infrared spectroscopy characterization (FTIR) of polymers with different reaction times is performed to prove the viewpoint (**Figure R2**). We observe the presence of alcoholic hydroxyl group during the reaction, indicating the production of side product (alcohol) and outgassing. Moreover, outgas is also measured via gas chromatography (GC). It is worth noting that outgassing also exists in other encapsulation strategies, which is an important issue to be addressed in the future.

Figure R2 FTIR spectra of the FDP at various times during the condensation reaction.

In the revised version, **Figure R2** is added as **Supplementary Figure 2b** and the relative discussion has been added on page 3 in the main text (also copied as below).

*“A small number of outgas was produced during the reaction (**Supplementary Fig. 2b**).” (Page 3 in the main text)*

3. For Table R1, it would be better to include another column of “overall processing time”. Please discuss that CFDP encapsulation process actually takes longer time.

Response: Thanks for your suggestion. We have added the column “overall processing time” in **Table R2**. The overall processing time of UV resin encapsulation is 7 min, containing the UV exposure time (2 min) and the curing time (5 min). The overall processing time of POE encapsulation is 15 min, which is the time of vacuum hot pressing. The overall processing time of CFDP encapsulation is 30 min, containing the stirring reaction time (5 min) and the curing time (25 min). Because CFDP encapsulation is a room temperature self-crosslinked strategy, the longer waiting time generally doesn't evolve in more processing cost.

Table R2. Performance, encapsulation process, and cost summary of perovskite encapsulant.

Type	Adhesion strength (Mpa)	Thermal conductivity ($\text{W m}^{-1} \text{K}^{-1}$)	Stability (damp heat test and thermal cycling test)	Process complexity	Overall processing time (min)	Cost ($\text{\$ m}^{-2}$)
UV resin	0.19	0.18	80%/75%	UV light	7	23.87
POE	0.56	0.30	95%/93%	Vacuum hot pressing	15	1.70
CFDP	0.28	0.35	98%/95%	Simple	30	1.91

In the revised version, **Table R2** is added as **Supplementary Table 8** and the relative discussion has been added on page 13 in the main text. The overall processing time has been added to **Supplementary Note 1** in supplementary information (also copied as below).

“To sum up, the room temperature nondestructive encapsulation strategy exhibits excellent device stability, simple encapsulation process, and low cost compared with other encapsulation strategies (Supplementary Table 8 and Supplementary Note 1), showing great commercialization advantages.” (Page 13 in the main text)

4. Any methods to quantify the outgas?

Response: Thanks for your question. We use gas chromatography (GC) to quantify the outgas according to the previous study (*Science 368, eaba2412 (2020)*). Before GC measurements, the encapsulants are cured in vials and heated at 85 °C for 1 h to facilitate outgassing. After 1 h, the sealed gas was sampled by a heated glass 1 mL syringe. We use the GC to detect the outgassing vapors and quantify the peak intensity by peak area via the software. Further, the quantitative analysis is carried out by the peak area of the standard gases. The outgassing release of the CFDP is 11.5 $\mu\text{mol/g}$ during the reaction. For UV resin, it is difficult to accurately analyze the amount of outgas due to the unknown structure of the UV resin (commercial products) and the complex outgassing components. Moreover, except for the outgassing of the encapsulant, the perovskite may also outgas (For example, CH_3I , CH_3Br , and NH_3 , etc.) during the encapsulation process due to ultraviolet radiation or high-temperature hot pressing. Therefore,

outgassing is an important issue to be addressed in the future.

In the revised version, the relative discussion has been added on page 4 in the supplementary information (also copied as below).

“The gas chromatography (Shimadzu GC-2014C, molecular sieve-5A, and Ar as the carrier gas) was used to acquire the outgassing release during the reaction process. The encapsulant was cured in vials and heated at 85 °C for 1 h to facilitate outgassing. After 1 h, the sealed gas was sampled by a heated glass 1 mL syringe. We quantified the peak intensity by peak area via the software and calculated the amount of outgassing (11.5 $\mu\text{mol/g}$) by comparing the peak area of the standard gas.”

(Page 4 in the supplementary information)

5. It is confusing that Figure 5's JV curve shows the spiro-based device (5c) while stability result is based on PTAA-based device (5d). Please replace the JV in Figure 5c with PTAA-based device result.

Response: Thanks for your suggestion. We have replaced the *J-V* curves in **Figure 5c** with PTAA-based device result.

Figure R3 The *J-V* curves of MAPbI₃ perovskite-based normal devices before and after target encapsulation.

Reviewer #3:

The authors have addressed my previous concerns. The current version can be accepted.

Response: We deeply appreciate the reviewing efforts and positive comments.

REVIEWERS' COMMENTS

Reviewer #2 (Remarks to the Author):

The author has addressed all of the comments sufficiently.